# Semi-Supervised Learning for Molecular Graphs via Ensemble Consensus

**Rasmus Tirsgaard** [1]   **Laurits Fredsgaard** [* 1]   **Marisa Wodrich** [* 1]   **Mikkel Jordahn** [1]   **Mikkel N. Schmidt** [1]

## Abstract

Machine learning is transforming molecular sciences by accelerating property prediction, simulation, and the discovery of new molecules and materials. Acquiring labeled data in these domains is often costly and time-consuming, whereas large collections of unlabeled molecular data are readily available. Standard semi-supervised learning methods often rely on label-preserving augmentations, which are challenging to design in the molecular domain, where minor changes can drastically alter properties. In this work, we show that semi-supervised methods that rely on an ensemble consensus can boost predictive accuracy across a diverse range of molecular datasets, task types, and graph neural network architectures. We find that training with an ensemble consensus objective increases robustness in models and exhibits an effect similar to knowledge distillation; an individual member of an ensemble trained this way outperforms a full ensemble trained in a traditional supervised fashion in almost all cases. In addition, this type of semi-supervised training reduces calibration error.

## 1. Introduction

In recent years, machine learning has emerged as a transformative tool in the molecular sciences, accelerating discovery in areas ranging from predicting quantum mechanical properties (Schütt et al., 2021; 2017; Musaelian et al., 2023; Wood et al., 2025) to discovering novel drugs (Wong et al., 2024; Kellenberger et al., 2007; Vidler et al., 2013; Zhuang et al., 2014; Ren et al., 2023) and catalysts (Pillai et al., 2023; Sun et al., 2024; Bai et al., 2025). However, despite recent efforts to curate large labeled datasets (Merchant et al., 2023; Levine et al., 2026), the scarcity of labeled data remains a fundamental bottleneck.

In materials and drug discovery, labels often come from computationally expensive simulations, such as density functional theory (DFT), or resource-intensive laboratory measurements. Consequently, datasets with specialized high-quality labels are typically small, while large databases of unlabeled molecules (e.g., ZINC (Irwin et al., 2012; Kim et al., 2024)) are not fully exploited. The combination of abundant unlabeled data and scarce labeled data presents an ideal setting for semi-supervised learning (SSL).

Yet, many state-of-the-art methods are poorly suited for the molecular domain. Dominant techniques such as consistency training (Berthelot et al., 2019; Sohn et al., 2020) critically depend on data augmentation strategies that create perturbed copies of an input while preserving its label. Such augmentations are notoriously difficult to design for molecules, where minor structural changes can drastically alter the chemical properties we aim to predict. Meanwhile, approaches such as iterative pseudo-labeling (Scudder, 1965; Riloff & Wiebe, 2003; Huang et al., 2022) hinges on the ability to reliably rank predictions by confidence in order to select the best candidates for pseudo-labeling and to avoid reinforcing model errors. This highlights a critical gap where standard SSL benchmarks and algorithms do not translate well to the practical challenges of molecular science.

In this work, we propose an SSL method that does not require explicit data augmentations, but rather relies on an *ensemble consistency loss*. Specifically, we train a model ensemble where each member learns from labeled data using a standard supervised loss and from unlabeled data using a loss that promotes agreement among the ensemble members. While ensemble coupling in SSL has been explored previously (Sajjadi et al., 2016; Tarvainen & Valpola, 2017; Platanios, 2018), our formulation is theoretically grounded in an ensemble loss ambiguity decomposition, requires only a single training run, yields individual models with ensemble-level performance, and leads to more robust models. Our work makes five core contributions:

1. We provide a theoretical motivation for our approach based on the formal decomposition of ensemble error, which justifies the consensus as a high-quality, better-than-average supervisory signal.

---
[*]Equal contribution   [1]Cognitive Systems, Department of Applied Mathematics and Computer Science, Technical University of Denmark. Correspondence to: Rasmus Tirsgaard <rhti@dtu.dk>.

*Proceedings of the 43rd International Conference on Machine Learning*, Seoul, South Korea. PMLR 306, 2026. Copyright 2026 by the author(s).

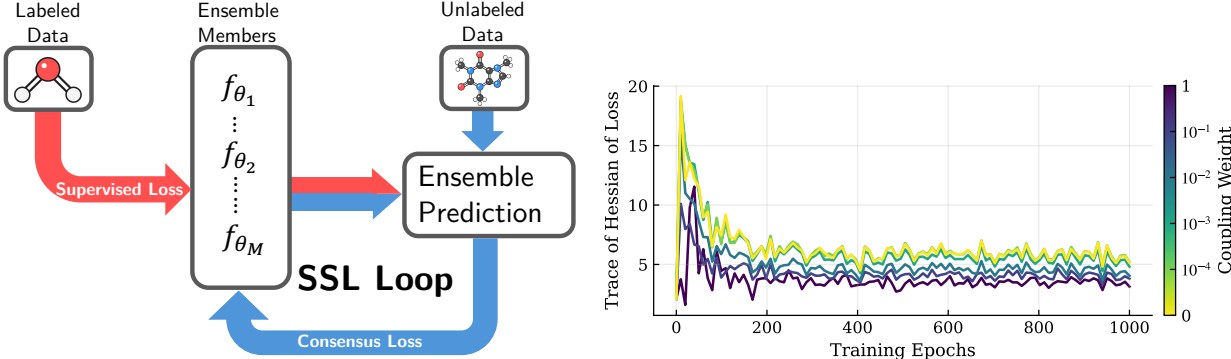

*Figure 1.* **(Left)** Schematic of the semi-supervised ensemble training loop. Individual ensemble members $f_\theta$ are trained on labeled data via a standard supervised loss. Simultaneously, the full ensemble's aggregated prediction on unlabeled data acts as a target for a consensus loss. Since the ensemble aggregate typically outperforms individual members, this creates a self-reinforcing loop where the collective prediction guides individual improvement. **(Right)** Training with ensemble consensus (provably) makes the model converge to more flat points in the loss landscape. This is shown here on the dataset QM9 target ZPVE as measured here by the trace of the Hessian of the loss on validation data.

2. We demonstrate that our method robustly improves predictive accuracy across a wide range of molecular datasets and architectures for both regression and classification.

3. We observe that the performance gap between individual members and the full ensemble is effectively eliminated. A single member of our consensus-trained ensemble even consistently outperforms a full baseline ensemble trained with traditional supervision.

4. We show that the ensemble-consensus objective also has a regularising effect, pushing the ensemble members towards more robust models.

5. We demonstrate that our model generally reduces calibration error compared to other SSL methods and maintains high predictive accuracy on the unlabeled training data.

## 2. Background

**Semi-Supervised Learning** Semi-supervised learning (SSL) is a machine learning paradigm designed for settings with a small amount of labeled data and a much larger amount of unlabeled data. The idea is to leverage the unlabeled data to learn about the underlying structure of the data distribution $p(x)$, which in turn improves the model's ability to learn the mapping from inputs to outputs, $p(y|x)$. Effective SSL methods are typically built upon one or more of the following assumptions:

- **Smoothness Assumption:** If two points $x_1, x_2$ are close in a high-density region of the underlying data manifold, their corresponding labels $y_1, y_2$ should also be close or identical.

- **Cluster Assumption:** The data tends to form distinct clusters, and points within the same cluster are likely to share the same label. This implies that a good decision boundary should lie in the low-density region between clusters.

**Consistency Loss** Consistency regularization is currently the most dominant family of SSL methods. The core idea is that the model's prediction for an unlabeled data point should remain consistent under small perturbations. This directly enforces the smoothness assumption. A successful perturbation or data augmentation is one that explores the local neighborhood of a data point on the manifold without changing its label. The objective is typically formulated as minimizing a distance measure (e.g., Mean Squared Error or KL-Divergence) between the model's predictions for two different augmentations of the same input:

$$\mathcal{L}_{\text{consistency}} = \mathbb{E}_{x_u \sim X_u}[D(f_\theta(\text{aug}_1(x_u))||f_\theta(\text{aug}_2(x_u)))].$$

Different choices of the perturbations give rise to a wide range of methods. Π-models (Sajjadi et al., 2016) enforce that two predictions should be the same under transformations to the data, with dropout, and random pooling providing perturbations to the model. Each unlabeled datapoint is passed through the network twice and penalized for the difference in the predictions between the passes. The benefit of consistency loss is highly linked to the quality of the data augmentation techniques, as shown in (Xie et al., 2020). Temporal ensembling (Laine & Aila, 2017) builds upon this by maintaining an exponential moving average of predictions for each unlabeled example to create a more stable consistency target. Instead of applying a temporal averaging over the predictions, the mean-teacher method (Tarvainen & Valpola, 2017) averages the model weights and uses the

predictions of that model as the consistency target. In the above works, the predictions can be seen as coming from a sort of pseudo-ensemble. As the members of this pseudo-ensemble are based on the trajectory or perturbation of a single network, the diversity of the predictions is reduced and biased, which reduces the prediction accuracy as we later highlight.

This problem can be mitigated by introducing multiple different initial weightings of the same architecture and training them in parallel to use as consistency targets. (Chen et al., 2021) (cross pseudo supervision) proposes to do this for pixel-wise segmentation, where the prediction of each of the two ensemble members is hard labeled and used as the consistency target. (Filipiak et al., 2021) further extends this for pixel-wise segmentation by using $n$ ensemble models and taking all combinations of hard labeled predictions as the consistency targets. Another paper that explores different ensemble predictions is (Platanios, 2018). Here the ensemble members are restarted multiple times during training, and the consensus target is computed from a trainable majority vote or Restricted Boltzmann Machine. All the above methods can be seen as stemming from a broad class of SSL methods that rely on the prediction of an ensemble to guide the training of the individual models to improve predictive accuracy.

In many applications, there exist few or no data augmentations that preserve the label of a data point. Examples include molecules, where the chemical properties can be changed significantly under small changes to the molecule. This restricts the consistency loss methods to only rely on perturbations to the model and not the data. This makes the class of ensemble-based SSL methods well-suited for the problem.

**Pseudo-labeling**    Pseudo-labeling (Yarowsky, 1995; Scudder, 1965; Riloff & Wiebe, 2003), also known as self-training or entropy minimization, is a process where an initial model is trained on the labeled data points and then used to predict labels for a large unlabeled dataset. The primary risk of this method is confirmation bias: if the model generates an incorrect pseudo-label with high confidence, it will reinforce its own mistake during retraining, leading to error propagation. To mitigate this risk, modern SSL methods often integrate more sophisticated frameworks. For example, one uncertainty-aware approach uses a model's evidential uncertainty to estimate the quality of each pseudo-label. This enables an adaptive weighting scheme where high-uncertainty (low-quality) pseudo-labels are given a smaller weight in the loss function, reducing their biasing effect. While this can be effective, such a strategy requires an initial, full training phase on the labeled data before the episodic pseudo-labeling can begin. It also introduces several additional tunable hyperparameters related to its

episodic schedule, which require careful tuning (Huang et al., 2022).

**Knowledge Distillation**    Knowledge distillation (Bucila et al., 2006; Hinton et al., 2015) was proposed as a way of using a complex "teacher" model to transfer its knowledge to a simpler "student" model. Usually, the teacher model is either a model with more parameters or the same model with multiple predictions averaged over multiple augmentations of the input, but the use of an ensemble as the teacher has also been explored (Hinton et al., 2015; Fukuda et al., 2017; Malinin et al., 2020). The transfer of knowledge can be enforced at different levels, such as feature representations (Heo et al., 2019) or intermediate layers (Zagoruyko & Komodakis, 2017). Approaches that match predictions are most closely related to our work. Aligning student and teacher predictions resembles the use of consistency targets in semi-supervised learning, with the key distinction that distillation is typically applied post-hoc, and thus lacks a bootstrapping effect where the teacher also benefits from the student's progress. Furthermore, knowledge distillation is often focused on preserving the uncertainty calibration of the teacher or achieving computational efficiency by deploying the smaller student model instead of the larger one.

## 3. Theoretical Motivation

The theoretical motivation for our method is grounded in the formal relationship between an ensemble's performance and that of its individual members. Ensemble performance is governed by a fundamental trade-off between the accuracy of the individual models and the diversity of their predictions. This relationship can be expressed through a loss decomposition, which shows that for any convex loss function, the ensemble's loss is guaranteed to be less than or equal to the average of the individual losses (Wood et al., 2023). This stems from Jensen's inequality and takes the general form:

$$\text{Ensemble Loss} = \text{Average Individual Loss} - \text{Ambiguity} \tag{1}$$

The ambiguity (or diversity) term is a non-negative quantity measuring disagreement among the members. This decomposition reveals that optimal ensemble performance requires not only accurate individual models but also beneficial diversity.

**Mean Squared Error**    This principle is most clearly illustrated in regression with Mean Squared Error (MSE), where the decomposition is exact and well-established (Krogh & Vedelsby, 1994). For an ensemble of $M$ models $\{f_{\theta_m}\}_{m=1}^{M}$ with a mean prediction $\bar{f}(x)$, the decomposition is:

$$\overbrace{(y - \bar{f}(x))^2}^{\text{Ensemble MSE}} \qquad (2)$$

$$= \underbrace{\frac{1}{M} \sum_{m=1}^{M} (y - f_m(x))^2}_{\text{Average Individual MSE}} - \underbrace{\frac{1}{M} \sum_{m=1}^{M} (\bar{f}(x) - f_m(x))^2}_{\text{Ambiguity (Prediction Variance)}}.$$

Here, the ambiguity is simply the variance of the predictions around the ensemble mean, providing a clear, label-independent measure of diversity.

**Cross-Entropy** The same principle extends to classification, though the decomposition for Cross-Entropy (CE) loss is more nuanced. Using the geometric mean to average probabilities across the ensemble yields a clean, label-independent decomposition, as in regression (Wood et al., 2023). An exact decomposition is also available for the arithmetic mean:

$$\overbrace{-\mathbf{y} \cdot \ln \bar{\mathbf{f}}}^{\text{Ensemble CE Loss}} \qquad (3)$$

$$= \underbrace{-\frac{1}{M} \sum_{m=1}^{M} \mathbf{y} \cdot \ln \mathbf{f}_m}_{\text{Avg. Individual CE Loss}} - \underbrace{\sum_{c=1}^{C} y_c \ln \frac{\frac{1}{M} \sum_{m=1}^{M} f_{m,c}}{(\prod_{m=1}^{M} f_{m,c}^{1/M})}}_{\text{Ambiguity (Label-Dependent)}},$$

although here the ambiguity term is explicitly a function of the true label vector $\mathbf{y}$ (where $y_c$ is the true probability of class $c$), making it label-dependent (Wood et al., 2023). Crucially, this ambiguity term is still guaranteed to be non-negative, ensuring that the ensemble loss is always less than or equal to the average individual loss.

Because the ensemble consensus is provably superior to the average individual model, using it as a consistency target for unlabeled data is both effective and theoretically well-justified. In addition, the ensemble prediction will be a useful signal as long as the models are better than random. This suggests the ensemble prediction does not need to incorporate a warm-startup to provide a useful predictive signal, as other works have observed (Tarvainen & Valpola, 2017) and used (Filipiak et al., 2021; Platanios, 2018).

## 4. Method

### 4.1. Formal Description

We address a standard semi-supervised learning problem with a small set of labeled data, $\mathcal{D}_L = \{(x_i, y_i)\}_{i=1}^{N_L}$, and a large set of unlabeled data, $\mathcal{D}_U = \{u_j\}_{j=1}^{N_U}$. We assume that both datasets are drawn from the same underlying distribution. Our method utilizes a deep ensemble of $M$ models, $\bar{f} = \{f_{\theta_m}\}_{m=1}^{M}$, initialized with different random weights. The training objective is defined on each model $f_{\theta_m}$ within

the ensemble. At each training step, its parameters $\theta_m$ are updated to minimize a composite loss, $\mathcal{L}_m$. The loss combines a standard supervised signal $\mathcal{L}_{\text{sup}}$ with an ensemble-driven consistency signal $\mathcal{L}_{\text{consistency}}$ that couple the ensemble members together:

$$\mathcal{L}_m = \mathcal{L}_{\text{sup}}(f_{\theta_m}, B_L) + \gamma \mathcal{L}_{\text{consistency}}(f_{\theta_m}, \bar{f}, B_U), \quad (4)$$

where $B_L$ and $B_U$ are mini-batches of labeled and unlabeled data, respectively, and $\gamma$ is the coupling weight. During training, all models are updated simultaneously by minimizing the sum of their individual losses i.e. $\mathcal{L} = \sum_{m=1}^{M} \mathcal{L}_m$. The first term, $\mathcal{L}_{\text{sup}}$, is the standard task-specific loss for model $f_{\theta_m}$ on the labeled batch, such as mean squared error (MSE) for regression or cross-entropy (CE) for classification. The second term, $\mathcal{L}_{\text{consistency}}$, provides the semi-supervised signal. It is calculated for the model $f_{\theta_m}$ but depends on the outputs of the entire ensemble. For each unlabeled sample $u \in B_U$, a consensus prediction, $\bar{f}(u)$, is computed by averaging the predictions of all $M$ models:

$$\bar{f}(u) = \frac{1}{M} \sum_{m=1}^{M} f_{\theta_m}(u). \qquad (5)$$

The consensus prediction serves as the augmentation-free consistency target for model $f_{\theta_m}$. We penalize the discrepancy between model prediction and the ensemble consensus as

$$\mathcal{L}_{\text{consistency}}(f_{\theta_m}, \bar{f}, B_U) = \frac{1}{|B_U|} \sum_{u \in B_U} D\left(f_{\theta_m}(u), \bar{f}(u)\right). \qquad (6)$$

Here, $D$ is a suitable distance metric, for example, the task-specific supervised loss (e.g., L2 or KL-divergence). In practice, when minimizing the loss we detach the gradient through $\bar{f}(u)$, as the consensus prediction is observed to at least as accurate as the individual members' predictions on average (see Section 3), ensuring that the ensemble is not encouraged to match the less accurate individual predictions. Note, not detaching the gradient has been observed to result in failure cases such as *learner collusion* (Jeffares et al., 2023), but in our experience it does not appear to affect results negatively, see Appendix E.8.

### 4.2. Consensus–Diversity Dynamics

Our proposed SSL training scheme directly manipulates the trade-off between accurate individual models and high diversity among them. The unsupervised loss term, $\mathcal{L}_{\text{consistency}}(x_u) = \mathcal{L}(f_{\theta_i}(x_u), \bar{f}(x_u))$, creates a pull towards consensus by guiding each model $f_{\theta_i}$ to agree with the more stable ensemble prediction $\bar{f}$. This directly reduces the average individual error by providing a high-quality supervisory signal for unlabeled data.

Simultaneously, this pull is counteracted by forces that preserve diversity. Each model begins from a unique random initialization and follows a distinct optimization path due to the stochastic nature of mini-batch SGD. This dynamic allows the models to converge to different solutions in parameter space while still agreeing in function space.

Therefore, our method does not eliminate diversity but rather regulates it. The hyperparameter $\gamma$ in the total loss $\mathcal{L} = \mathcal{L}_l + \gamma \mathcal{L}_{\text{consistency}}$ serves as a direct control over this balance, allowing us to leverage the unlabeled data to improve individual model accuracy without forcing a complete collapse in diversity. See appendix B.5 for details.

Another benefit of this continuous learning between models is that the models should be less likely to get stuck on early bad predictions, as can be the case with many forms of pseudo-labeling. This is because the ensemble targets are not frozen but "moves" with the ensemble. This can explain why we observe no benefit to warmup of the coupling loss.

### 4.3. Relationship to Bayesian Posteriors

If we treat one model trained from a classic deep learning as a posterior sample, the ensemble prediction converges to the posterior mean as the number of ensemble members increases towards infinity. This principle guides ensemble distillation to let an individual model approximate the posterior mean. In appendix A, we show that consensus training also converges to the posterior mean, but only for an ensemble with linear models. Instead, for non-linear models there is an additional term that biases members in the ensemble towards the least parameter-sensitive predictions in the ensemble. This is generally a benevolent effect (Hochreiter & Schmidhuber, 1997; Kaddour et al., 2022; Keskar et al., 2017; Andriushchenko et al., 2023), and we can understand it as the members will learn to prioritize the simplest explanation all members can agree on. We hypothesize this is the main reason individual models in the ensemble consensus can outperform a full standard ensemble.

## 5. Experimental setup

We evaluate our method in two settings: First, on a quantum chemistry benchmark to demonstrate its relevance for 3D-geometry-based molecular property prediction, and then across a diverse suite of graph-level tasks to assess its broader applicability. All ensemble members were trained on identical mini-batches of supervised data to simplify implementation. While this strategy reduces ensemble diversity, potentially limiting the ensemble's predictive power, it allows for a fair direct comparison with single models. The code is available here https://github.com/lauri tsf/semi-supervised-ensemble-training.

**Semi-supervised Protocol** To simulate the common scenario of data scarcity, we restrict the supervised portion of our training to a small fraction for each task (10%). The remaining training data (90%) is treated as unlabeled and is used exclusively for our ensemble consistency loss. Our primary baseline is a standard deep ensemble of the same architecture, trained only on this small labeled data subset. This setup allows us to directly measure the performance gain from leveraging unlabeled data.

**Datasets** We test our method on a wide range of different datasets. We perform a prediction of molecular properties in the QM9 dataset (Wu et al., 2018) for the main 12 targets, using the PaiNN architecture (Schütt et al., 2021) and an ensemble of size $M = 4$. To investigate how our method scales, we study and compare performance on a single target (internal energy at 0K) for different ensemble sizes ($M \in \{1, 2, 3, 4\}$). We also study the performance on harder data splits in Appendix B.6. For broader validation of our method, we adopt a comprehensive benchmark suite of graph-level tasks. We use three different graph-based architectures: GCN (Kipf & Welling, 2017), GIN (Xu et al., 2019), and GatedGCN (Bresson & Laurent, 2017), adapting the code from (Luo et al., 2025) and following the testing procedure from (Rampásek et al., 2022). We refer to this suite of benchmarks as GNN+ benchmarks. The ensemble size is fixed to $M = 4$. To demonstrate the general applicability of our method beyond the molecular domain, we perform experiments on a benchmark of non-molecular graph datasets (see Appendix B.3). Further experiments showing broader applicability beyond graphs are included in Appendix B.1. All datasets were split into 80% training data, 10% validation data and 10% test data. The training data was further split into 10% labeled training data and the labels for the remaining 90% where discarded to used as unlabeled (unsupervised) training data.

**Hyperparameter Tuning** To ensure well-tuned models for datasets, the training hyperparameters (learning rate and weight decay) were optimized for each target and model based on the validation performance of a single model in the supervised setting on the reduced labeled data. These hyperparameters were kept fixed across different SSL methods tested to ensure fair comparison. The parameters associated with each specific SSL method (coupling weight, mean-teacher decay, etc.) were optimized based on validation accuracy for each target on QM9, and selected for the GNN+ datasets based on the best value of ZINC. Details about the tuning procedures and selected hyperparameters can be found in Appendix C.

**Evaluation** We evaluate the predictive performance for a single model, a standard ensemble, an ensemble using SSL via ensemble consensus (ours) and an individual

*Table 1.* PaiNN performance (MAE) on QM9 targets. Results are reported as mean $\pm 1.96$ standard error of the mean over 5 seeds. Background colors indicate performance relative to the supervised ensemble baseline: green denotes lower error (improvement), while red denotes higher error.

| | | Individual Member | | Ensemble (M=4) | | | |
| --- | --- | --- | --- | --- | --- | --- | --- |
| Target | Unit | Supervised | Supervised + SSL | Supervised | Supervised + SSL | Mean-teacher | PSEUD$\sigma$ |
| $\mu$ | D | $.0737_{\pm.0008}$ | $.0619_{\pm.0003}$ | $.0680_{\pm.0006}$ | $.0613_{\pm.0003}$ | $.0721_{\pm.0024}$ | $.06487_{\pm.00088}$ |
| $\alpha$ | $a_0^3$ | $.1622_{\pm.0011}$ | $.1322_{\pm.0011}$ | $.1419_{\pm.0009}$ | $.1303_{\pm.0011}$ | $.1569_{\pm.0020}$ | $.1454_{\pm.0004}$ |
| $\epsilon_{\text{HOMO}}$ | meV | $80.6061_{\pm.5062}$ | $73.9789_{\pm.4368}$ | $76.4713_{\pm.5361}$ | $73.0755_{\pm.4472}$ | $80.6090_{\pm2.0301}$ | $78.72_{\pm1.1196}$ |
| $\epsilon_{\text{LUMO}}$ | meV | $62.0372_{\pm.4253}$ | $57.7186_{\pm.2247}$ | $59.3140_{\pm.4609}$ | $57.2369_{\pm.2159}$ | $61.9719_{\pm.8156}$ | $59.74_{\pm0.6248}$ |
| $\Delta\epsilon$ | meV | $125.2102_{\pm.4734}$ | $117.0365_{\pm.4988}$ | $119.4206_{\pm.4468}$ | $115.7195_{\pm.5100}$ | $125.4843_{\pm2.5836}$ | $122.5_{\pm.7501}$ |
| $\langle R^2 \rangle$ | $a_0^2$ | $.7922_{\pm.0284}$ | $.6100_{\pm.0206}$ | $.6246_{\pm.0205}$ | $.5605_{\pm.0206}$ | $.7987_{\pm.04019}$ | $.9099_{\pm.0157}$ |
| ZPVE | meV | $2.2197_{\pm.0055}$ | $2.0138_{\pm.0054}$ | $2.0743_{\pm.0054}$ | $1.9907_{\pm.0055}$ | $2.1821_{\pm.0380}$ | $2.141_{\pm.0123}$ |
| $U_0$ | meV | $24.8752_{\pm.1477}$ | $19.9642_{\pm.1291}$ | $20.9101_{\pm.1557}$ | $19.3816_{\pm.1278}$ | $24.7288_{\pm.8505}$ | $24.00_{\pm.2929}$ |
| $U$ | meV | $25.1281_{\pm.2025}$ | $20.1731_{\pm.1577}$ | $21.0971_{\pm.1914}$ | $19.5886_{\pm.1574}$ | $24.9622_{\pm.9955}$ | $24.28_{\pm.4701}$ |
| $H$ | meV | $25.1240_{\pm.1981}$ | $20.1407_{\pm.1268}$ | $21.0892_{\pm.1946}$ | $19.5509_{\pm.1328}$ | $24.8495_{\pm.9604}$ | $24.33_{\pm.6181}$ |
| $G$ | meV | $25.3804_{\pm.1856}$ | $20.3142_{\pm.1571}$ | $21.4136_{\pm.1811}$ | $19.7479_{\pm.1634}$ | $25.1641_{\pm.7511}$ | $24.33_{\pm.4412}$ |
| $C_v$ | $\frac{\text{cal}}{\text{mol K}}$ | $.0567_{\pm.0004}$ | $.0449_{\pm.0002}$ | $.0489_{\pm.0004}$ | $.0439_{\pm.0002}$ | $.0556_{\pm.0002}$ | $.05408_{\pm.00027}$ |

member from the latter. In addition, we implement both mean-teacher (Tarvainen & Valpola, 2017), and a strong pseudo-label based approach that have been used on QM9, PSEUD$\sigma$ (Huang et al., 2022) . All results are reported as the mean along with 1.96 times the standard error of the mean across different seeds.

# 6. Results

## 6.1. Molecular Property Prediction on QM9

The performance of our method on the 12 regression targets of the QM9 dataset is presented in Table 1. The results indicate that training with the ensemble consistency loss ("Supervised + SSL") reduces the MAE across all evaluated targets when compared to the supervised-only baseline. This is observed for both the individual PaiNN models and the four-member ensembles. Furthermore, a single individual model from the coupled ensemble consistently outperforms the traditional supervised ensemble on all targets.

The results for molecular property prediction on QM9 for different ensemble sizes are shown in Table 2. Our SSL method outperforms a traditional ensemble for all sizes tested. Additionally, using an individual member from an ensemble trained using our proposed SSL method, we not only outperform a standard single model but also perform at a similar level to an ensemble that has only been trained on the supervised data. The results are consistent across all ensemble sizes. Performance increases with more ensemble members.

## 6.2. GNN+ Benchmark

To assess the broader applicability of our method, we evaluate it on several molecule-related benchmarks using three

*Table 2.* PaiNN performance (MAE) on QM9 internal energy at 0K in eV ($U_0$) for different ensemble sizes averaged across 5 seeds, with mean $\pm 1.96$ standard error of the mean.

| | Individual member | | Ensemble | |
| --- | --- | --- | --- | --- |
| Size (M) | Supervised | Supervised + SSL | Supervised | Supervised + SSL |
| 1 | $24.88_{\pm0.15}$ | – | – | – |
| 2 | – | $20.89_{\pm0.30}$ | $22.18_{\pm0.44}$ | $20.43_{\pm0.29}$ |
| 3 | – | $20.44_{\pm0.16}$ | $21.31_{\pm0.23}$ | $19.93_{\pm0.17}$ |
| 4 | – | $19.96_{\pm0.13}$ | $20.91_{\pm0.16}$ | $19.38_{\pm0.13}$ |

different GNN architectures. The results are summarized in Table 3, and are consistent with the performance on QM9, even when the coupling weight is kept the same across datasets. Looking at a single model, the addition of the SSL task consistently improves performance over the supervised-only baseline across all datasets and architectures. This performance gain also translates to the full ensembles, which show improvement when trained with the consistency loss. The performance of a single model trained with our SSL method often exceeds that of an entire ensemble trained only on labeled data.

## 6.3. Computational Overhead

Increasing the ensemble size reduces the error but also increases the computational overhead. We investigate this relationship for training and inference time on QM9 target ZPVE using the same setup as previous QM9 experiments. One epoch here is defined to mirror our QM9 Experiments, i.e. the full QM9 supervised data, while the SSL models also iterates over an equal amount of unlabeled batches. The results are shown in Figure 2. From the error vs inference, we see how a single model from the SSL ensemble per-

*Table 3.* Performance on molecule-related benchmarks using different GNN architectures averaged across 5 seeds using the optimal coupling weight found on ZINC with GCN. Pairwise loss is defined in Appendix E.2. Background colors represent the deviation from the supervised ensemble baseline for each architecture: green denotes lower error (improvement), while red denotes higher error.

| Dataset | Training | Metric | GCN | | GIN | | GatedGCN | |
|---|---|---|---|---|---|---|---|---|
| | | | Individual | Ensemble | Individual | Ensemble | Individual | Ensemble |
| ZINC | Supervised | MAE ↓ | $.3163_{\pm.0121}$ | $.2934_{\pm.0094}$ | $.2765_{\pm.0247}$ | $.2516_{\pm.0136}$ | $.2920_{\pm.0113}$ | $.2646_{\pm.0235}$ |
| | Consensus | | $.2406_{\pm.0150}$ | $.2367_{\pm.0148}$ | $.2519_{\pm.0246}$ | $.2485_{\pm.0232}$ | $.2717_{\pm.0230}$ | $.2658_{\pm.0177}$ |
| | Pairwise | | $.2462_{\pm.0108}$ | $.2390_{\pm.0102}$ | $.2500_{\pm.0083}$ | $.2462_{\pm.0092}$ | $.2653_{\pm.0158}$ | $.2597_{\pm.0171}$ |
| | Mean teacher | | $.2884_{\pm.0128}$ | – | $.2791_{\pm.0117}$ | – | $.2830_{\pm.0159}$ | – |
| Peptides-struct | Supervised | MAE ↓ | $.3047_{\pm.0098}$ | $.2932_{\pm.0084}$ | $.2966_{\pm.0067}$ | $.2918_{\pm.0058}$ | $.2994_{\pm.0105}$ | $.2908_{\pm.0101}$ |
| | Consensus | | $.2868_{\pm.0062}$ | $.2866_{\pm.0061}$ | $.2944_{\pm.0072}$ | $.2938_{\pm.0068}$ | $.2854_{\pm.0061}$ | $.2848_{\pm.0068}$ |
| | Pairwise | | $.2933_{\pm.0031}$ | $.2892_{\pm.0029}$ | $.2916_{\pm.0030}$ | $.2901_{\pm.0029}$ | $.2898_{\pm.0042}$ | $.2870_{\pm.0041}$ |
| | Mean teacher | | $.2985_{\pm.0029}$ | – | $.2948_{\pm.0023}$ | – | $.2953_{\pm.0034}$ | – |
| Peptides-func | Supervised | AP ↑ | $.4931_{\pm.0346}$ | $.5105_{\pm.0342}$ | $.4566_{\pm.0224}$ | $.4765_{\pm.0327}$ | $.4289_{\pm.0051}$ | $.4444_{\pm.0200}$ |
| | Consensus | | $.5070_{\pm.0141}$ | $.5160_{\pm.0141}$ | $.4756_{\pm.0180}$ | $.4815_{\pm.0179}$ | $.4509_{\pm.0144}$ | $.4580_{\pm.0062}$ |
| | Pairwise | | $.5055_{\pm.0151}$ | $.5163_{\pm.0150}$ | $.4739_{\pm.0110}$ | $.4811_{\pm.0117}$ | $.4463_{\pm.0067}$ | $.4548_{\pm.0069}$ |
| | Mean teacher | | $.4893_{\pm.0169}$ | – | $.4611_{\pm.0130}$ | – | $.4352_{\pm.0058}$ | – |
| ogbg-molhiv | Supervised | AUROC ↑ | $.7216_{\pm.0193}$ | $.7357_{\pm.0212}$ | $.7329_{\pm.0166}$ | $.7346_{\pm.0165}$ | $.7312_{\pm.0081}$ | $.7341_{\pm.0107}$ |
| | Consensus | | $.7308_{\pm.0218}$ | $.7357_{\pm.0212}$ | $.7339_{\pm.0149}$ | $.7347_{\pm.0153}$ | $.7361_{\pm.0069}$ | $.7383_{\pm.0073}$ |
| | Pairwise | | $.7247_{\pm.0160}$ | $.7336_{\pm.0146}$ | $.7273_{\pm.0128}$ | $.7294_{\pm.0128}$ | $.7375_{\pm.0052}$ | $.7403_{\pm.0050}$ |
| | Mean teacher | | $.7213_{\pm.0161}$ | – | $.6996_{\pm.0207}$ | – | $.7295_{\pm.0165}$ | – |
| ogbg-molpcba | Supervised | AP ↑ | $.1368_{\pm.0025}$ | $.1578_{\pm.0030}$ | $.1421_{\pm.0026}$ | $.1567_{\pm.0029}$ | $.1615_{\pm.0034}$ | $.1779_{\pm.0043}$ |
| | Consensus | | $.1476_{\pm.0023}$ | $.1585_{\pm.0026}$ | $.1496_{\pm.0033}$ | $.1567_{\pm.0039}$ | $.1701_{\pm.0036}$ | $.1781_{\pm.0034}$ |
| | Pairwise | | $.1471_{\pm.0027}$ | $.1597_{\pm.0028}$ | $.1498_{\pm.0021}$ | $.1574_{\pm.0024}$ | $.1674_{\pm.0032}$ | $.1765_{\pm.0034}$ |
| | Mean teacher | | $.1435_{\pm.0016}$ | – | $.1479_{\pm.0037}$ | – | $.1669_{\pm.0028}$ | – |

forms better than a full normal ensemble at a fraction of the inference time. The hyperparameters used for computing the graph was a batch size of 32 for training and 256 for inference speed with warmup, all on a single NVIDIA RTX A5000, and averaged over 10 epochs or 50 batches respectively. The experiments were performed with Pytorch version 2.7.1 and CUDA 11.8.

## 6.4. Variance of the Ensemble

Intuitively, the strength of coupling impacts the diversity of the ensemble. Using the same setup as previous QM9 experiments on target ZPVE, we investigate this in Figure 3. We see that there still is still diversity in the predictions on validation data even for high coupling strengths, but it decreases with coupling strength. This underlines that the ensemble members are able to distil the generalised behavior of the full ensemble during training. Interestingly, the diversity on the training data actually increases initially with coupling strength. We suspect this is due to the reguralising effect that result in more robust models being learned. For completion, we also plot the variance within the ensemble over epochs and for unsupervised data in Appendix B.5

## 6.5. Unsupervised Learning

We also investigate how the coupled ensembles behave with data from another distribution, specifically PCQM4Mv2 (Hu et al., 2021) and unsupervised methods. Specifically, we compare against Frad (Feng et al., 2023), which is an unsupervised model specifically designed for 3D-molecular datasets such as QM9. Frad works by denoising 3D positions of molecules, and can be used two different ways. Firstly to pre-train on a unlabeled dataset, referred to here as pre-trained. Secondly, as a unsupervised objective defined on both labeled and unlabeled data and optimised jointly with the supervised signal, noted here as denoising. The experiments are conducted using the dataset PCQM4Mv2 as the unlabeled data and all labels from the training set of QM9. We reuse the architecture, hyperparameters, and code from the official implementation of Frad. Notably, the architecture of the models is changed from PaiNN to MDNet (Thölke & Fabritiis, 2022). To make comparisons more fair computationally, we set the ensemble size to be 2. After an initial sweep over ensemble consensus coupling weights of $(10^{-2}, 10^{-3}, 10^{-4}, 10^{-5}, 10^{-6})$, we found that $\gamma = 10^{-5}$ was optimal. The results of these experiments are shown in Table 4. Each model in the ensemble is initialed from a model that was pre-trained on the unsupervised data using different seeds for each ensemble member.

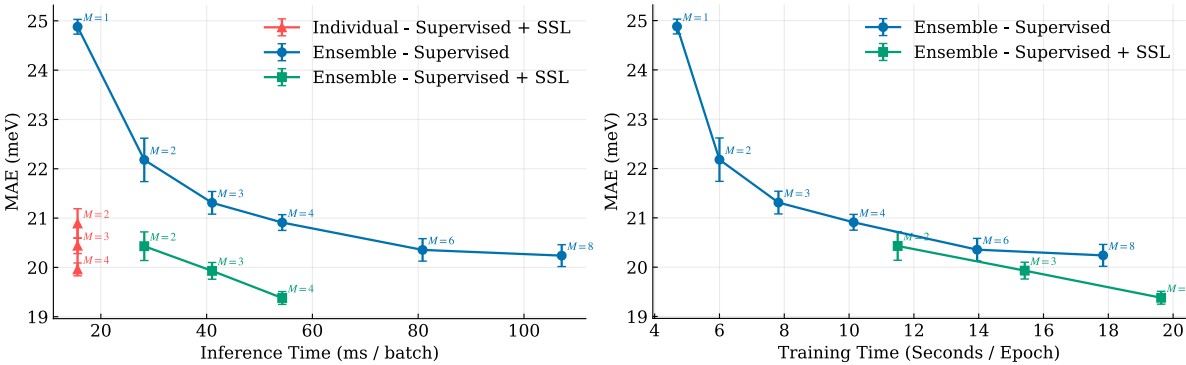

*Figure 2.* The validation error on QM9 for target ZPVE (target 7) as a function of (Left) inference time and (Right) training time. The results are averaged over 5 seeds, with a $95\%$ error of the mean visualised.

*Table 4.* Error in MAE on target $\epsilon_{\text{HOMO}}$ on QM9, when using PCQM4Mv2 as the unlabeled dataset.

| Pre-train | Denoise | Consensus | Individual | Ensemble |
|:---:|:---:|:---:|:---:|:---:|
| ✓ | | | 19.26 | 17.38 |
| ✓ | | ✓ | 17.01 | 16.75 |
| ✓ | ✓ | | 16.90 | 15.56 |
| ✓ | ✓ | ✓ | 15.49 | 14.54 |
| | ✓ | | 20.18 | 18.10 |
| | ✓ | ✓ | 19.98 | 17.01 |
| | | | 22.37 | 21.16 |
| | | ✓ | 20.75 | 19.87 |

We generally observe that consensus training reduces the error slightly, but that the 3D-molecular denoising objective from Frad yield larger gains. Interestingly, we do see that using consensus improved upon the pretrained models, suggesting that the coupled ensembles can still extract additional information from the unlabeled data. This shows that a modality-specific objective is still preferred, but ensemble consensus when computational resources allows. Notable, we did observe some training instabilities when the using consensus with pre-trained models. We hypothesize this is due to the consensus forcing the model predictions to be the same initially, which could result in the learned representations from pre-training being lost.

## 7. Discussion

Our experiments on QM9 and the more varied GNN+ benchmark show that our ensemble-based SSL framework consistently improves model predictive performance in low-data regimes. The most significant finding is the substantial boost in accuracy for individual models, a direct result of the knowledge transferred from the ensemble's consensus on unlabeled data. This finding is similar to the idea of ensemble distillation (Hinton et al., 2015), where the knowledge of an ensemble is transferred to a single, smaller model. A new effect observed for the dataset (QM9) with a tuned coupling weight, where the individual models obtain a bet-

ter predictive accuracy than the entire consensus ensemble. This can be seen as semi-supervised effect where the gradual improvement of a single model results in even better ensemble consensus targets for individual models to learn from. A possible explanation is the effect described in appendix A, where the models become more robust in their prediction. Specifically, we find that the Hessian of the validation loss decreases (Figure 1 (right)), and that the sensitivity of the prediction to changes (Appendix A.3) both decreases when increasing the coupling weight.

This has a key practical benefit: while the method requires an ensemble during training, a single, improved model can be deployed for inference. This offers a valuable trade-off, where an increased one-time training cost yields a final model that is both highly accurate and computationally efficient at inference time. Chemical property screening is a compelling use-case, as vast databases of molecules need to be screened, resulting in a high inference cost, while the available labeled data and models are small, making training cheap.

It is noteworthy that for datasets where the parameter related to SSL ($\gamma$ or the mean-teacher decay) was not directly tuned, the improvement in predictive accuracy was noticeably smaller. This indicates the SSL parameter is highly dependent on the specific dataset. This finding is also supported in Appendix E.4 and E.6, which shows that picking the best (constant) coupling weight is more impactful than exploring more sophisticated coupling strategies such as warm up of the consensus loss.

As shown in Table 2, the predictive performance scales with the number of members in the consensus ensemble. Individual models from the ensemble trained with our method consistently perform at a similar level to an entire traditional ensemble across all ensemble sizes. This finding is further supported in Appendix B.1 for up to 32 ensemble members, with different architectures and datasets. This benefit comes at the cost of roughly 2x slower training time

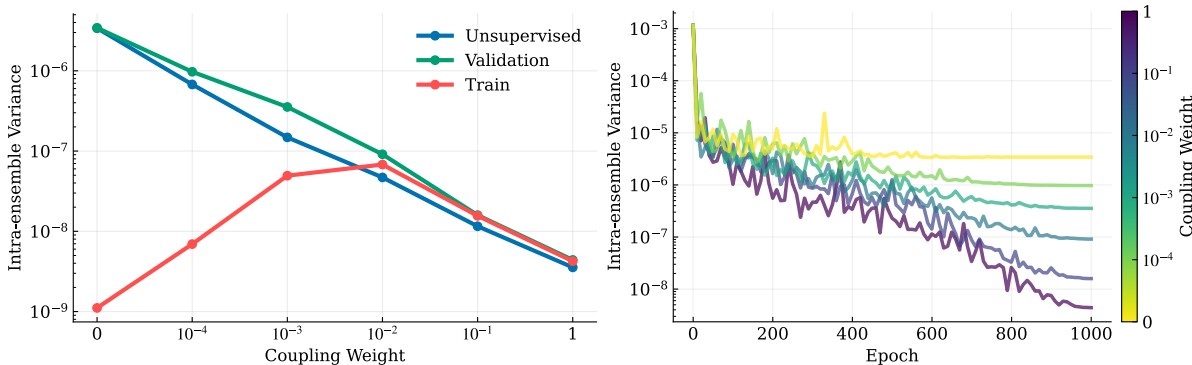

*Figure 3.* The variance within the ensemble across coupling weights on validation data (Left) and data splits versus training epochs (Right) for QM9 target ZPVE.

as detailed in Figure 2 as there simply is the additional unlabeled data to iterate over. Due to the ensemble members only communicating their predictions, the ensemble size can largely be mitigated by distributing the ensemble members across different devices and collecting predictions with ALL_GATHER/ALL_REDUCE, for example[1].

In Appendix D we see that the calibration of the individual ensemble members on ogbg-molhiv is improved by coupling. This indicates that the ensemble members also learns some of the uncertainty captured in the ensemble. This finding is further supported in Appendix B.1, where increasing the ensemble size consistently results in improved calibration for both the individual models and the full ensemble. The results for the calibration of the full ensemble is unclear. In ogbg-molhiv the calibration of the full coupled ensemble is generally better than the decoupled ensemble, but in Appendix B.1 both the ECE and Brier score suffers from coupling. It is not clear if this difference in findings is due to the different dataset and modality (graph versus image), or due to the coupling in Appendix B.1 not being chosen optimally.

We found that ensemble consensus can improve upon models pretrained or jointly trained with modality specific methods such as 3D positional denoising in molecules. While the improvement was small, it illustrates that ensemble consensus can be combined with other methods to improve model performance when the computational budget allows.

**Limitations**   The primary limitation of our approach is the computational overhead associated with training an ensemble consensus model. In addition to having to train $M$ identical models, the size of the training mini-batches is doubled, as it now includes both the unlabeled mini-batch in addition to the labeled mini-batch. This means the computation required roughly scales as $\mathcal{O}(2M)$. Observed timing can

be found in Appendix F. This overhead can be prohibitive for training large models, such as foundation models. Coupled ensembles are more practical in for smaller models or models trained on embeddings or latent representations.

**Future work**   Our findings suggest several promising avenues for future research. While this work created a semi-supervised split from a fully labeled dataset, a compelling next step would be to use all available labeled data for supervision while introducing a separate, truly unlabeled dataset. This would more directly quantify the benefit of leveraging vast, external chemical libraries and be of interest in a practical setting. Using ensembles for semi-supervised learning also opens the direction for improving accuracy in a principled manner by diversifying the ensemble members through existing techniques. Ideally in a way that can reduce the computation required. Furthermore, different strategies for how to couple an ensemble can be investigated. Our experiments (Appendix E.4) suggest that a constant coupling weight for ensemble consistency weight loss throughout the whole training yield the best results. Ideally, a more sophisticated weighting schedule could allow for coupling only in the later stages of the training to reduce the computational overhead.

## Impact Statement

This paper presents work that strives to advance the field of semi-supervised learning for especially molecular data. Machine learning is increasingly becoming a central tool for many chemical tasks. Of especial impact is it's increasing use in development of new medicine. We hope that the work presented here can help further improve the deep learning models used for such tasks to hopefully reduce the cost of development of medicine. Reducing the development cost should reduce the price of new medicine, making it more accessible.

---

[1]https://github.com/lauritsf/torch-distributed-ensemble

## Acknowledgements

The authors acknowledge support from the Novo Nordisk Foundation under grant no NNF22OC0076658 (Bayesian neural networks for molecular discovery).

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

# Supplementary material

# A. Theoretical Analysis

## A.1. Ensemble Consensus for Linear Models

**Theorem A.1.** *Assuming all ensemble members $f_{\theta_m}$ are linear models and the prediction $\bar{f}(x)$ of the ensemble is the minimizer of the sum of individual consensus losses, then $\bar{f}(x)$ stays constant during consensus updates.*

*Proof.* Let $\theta^m$ be the weights of the individual ensemble members and $\theta$ be the combined parameters of the ensemble. Under a consensus update, the change is

$$\theta_{t+1} = \theta_t - \eta \nabla_\theta \mathcal{L}(\bar{f}_{\theta_t}(x) - f_{\theta_t^m}(x)).$$

Meaning for some feature vector $\phi(x)$

$$\begin{aligned} f_{\theta_{t+1}}(x) &= \left(\theta_t - \eta \nabla_\theta \mathcal{L}(\bar{f}_{\theta_t}(x) - f_{\theta_t^m}(x))\right)^T \phi(x) \\ &= \theta_t^T \phi(x) - \eta \nabla_\theta \mathcal{L}(\bar{f}_{\theta_t}(x) - f_{\theta_t^m}(x))^T \phi(x) \\ &= f_{\theta_t}(x) - \eta \nabla_\theta \mathcal{L}(\bar{f}_{\theta_t}(x) - f_{\theta_t^m}(x))^T \phi(x). \end{aligned}$$

Now for the members respectively

$$f_{\theta_{t+1}^m}(x) = f_{\theta_t^m}(x) - \eta \nabla_{\theta^m} \mathcal{L}(\bar{f}_{\theta_t}(x) - f_{\theta_t^m}(x))^T \phi(x).$$

Since all the ensemble members are linear, it follows that $\nabla_{\theta^m} f_{\theta_t^m}(x) = \phi(x)$. This means:

$$\begin{aligned} &\bar{f}_{\theta_{t+1}}(x) - \bar{f}_{\theta_t}(x) \\ &= \frac{1}{M} \sum_{m=1}^{M} f_{\theta_t^m}(x) - \eta \frac{1}{M} \sum_{m=1}^{M} \nabla_{\theta^m} \mathcal{L}(\bar{f}_{\theta_t}(x) - f_{\theta_t^m}(x))^T \phi(x) - \frac{1}{M} \sum_{m=1}^{M} f_{\theta_t^m}(x) \\ &= \frac{-1}{M} \sum_{m=1}^{M} \eta \nabla_{\theta^m} \mathcal{L}(\bar{f}_{\theta_t}(x) - f_{\theta_t^m}(x))^T \phi(x) \\ &= -\frac{\eta}{M} \sum_{m=1}^{M} \nabla \mathcal{L}(\bar{f}_{\theta_t}(x) - f_{\theta_t^m}(x))^T \phi(x) \\ &= -\frac{\eta}{M} \left( \sum_{m=1}^{M} \nabla \mathcal{L}(\bar{f}_{\theta_t}(x) - f_{\theta_t^m}(x)) \right)^T \phi(x) \\ &= -\frac{\eta}{M} \underbrace{\left( \sum_{m=1}^{M} \nabla \mathcal{L}(\bar{f}_{\theta_t}(x) - f_{\theta_t^m}(x)) \right)^T}_{0=\text{ if } \bar{f}_{\theta_t}(x) \text{ minimizes the sum of losses}} \phi(x) \qquad (7) \\ &= 0. \end{aligned}$$

This above derivation relies on $\mathcal{L}$ being a proper loss function for the ensemble, i.e. that $\bar{f}_{\theta_t}(x)$, as it minimises the loss as by first order optimality conditions

$$\nabla_y \left[ \sum_{m=1}^{M} \mathcal{L}(y - f_{\theta_t^m}(x)) \right]_{y=\bar{f}_{\theta_t}(x)} = 0,$$

implying that

$$\sum_{m=1}^{M} \nabla_{\bar{f}_{\theta_t}} \mathcal{L}(\bar{f}_{\theta_t}(x) - f_{\theta_t^m}(x)) = 0.$$

Using the chain rule

$$\nabla_{\bar{f}_\theta} \mathcal{L}(\bar{f}_{\theta_t}(x) - f_{\theta_t^m}(x)) = -\nabla_{f_{\theta^m}} \mathcal{L}(\bar{f}_{\theta_t}(x) - f_{\theta_t^m}(x)),$$

we get

$$\sum_{m=1}^{M} \left[ \nabla_y \mathcal{L}(y - f_{\theta_t^m}(x)) \right]_{y=\bar{f}_{\theta_t}(x)} = \sum_{m=1}^{M} -\nabla_{f_{\theta^m}} \mathcal{L}(\bar{f}_{\theta_t}(x) - f_{\theta_t^m}) = 0.$$

$\square$

Note that the choice of loss function dictates what prediction the individual models converge to. The minimizer of MSE loss the arithmetic mean of the ensemble members, while for MAE, the ensemble prediction should be the median or all satisfy this condition. Under these conditions the ensemble prediction remains unchanged under purely consensus updates for linear models as the individual gradient contributions from the members cancels out.

**Lemma A.2.** *Assume the same conditions as Theorem A.1 and a coupled ensemble of linear models $\bar{f}_t^{coupled}$ optimised under gradient descent with the a joint consensus objective function i.e. Equation 4. Then the sequence of ensemble models from each step of optimisation are exactly the same for the coupled and a decoupled ensembles, i.e. $\bar{f}_t^{decoupled} = \bar{f}_t^{coupled}$, $\forall t$. As a result if the decoupled ensemble converge, the coupled ensemble converges to the same model.*

*Proof.* We will now prove that optimising the overall loss, i.e. jointly the supervised and unsupervised loss, the coupled ensembles converge to the same same as the decoupled ensembles for linear models. Let $\{\bar{f}_t^{decoupled}\}_{t=1}^n$ and $\{\bar{f}_t^{coupled}\}_{t=1}^n$, be a sequence from a decoupled and coupled ensemble over $t$ gradient steps. We will prove using induction that at each step of the equation that the coupled ensemble predictions $\{\bar{f}_t^{coupled}\}_{t=1}^n$ are equal to the decoupled ensemble, i.e. $\bar{f}_t^{decoupled} = \bar{f}_t^{coupled}$, $\forall t$. The individual models starts at with the same parameters, so $\bar{f}_0^{coupled} = \bar{f}_0^{decoupled}$. Using induction, we can assume for $t$ that $\bar{f}_t^{coupled} = \bar{f}_t^{decoupled}$. Since each member $f_{\theta^m}$ is linear, it follows that $\bar{f}_\theta$ is also linear. Hence, the update rule can be written as

$$\bar{f}_{\theta_{t+1}}(x) = \bar{f}_{\theta_t}(x) - \eta \nabla_\theta \mathcal{L}(\bar{f}_{\theta_t})^T \phi(x).$$

We now consider the joint optimization of the supervised and ensemble consensus objective. Since the joint objective is additive, we can split the gradient for update $t$ update as

$$\nabla_\theta \mathcal{L}(\bar{f}_{\theta_t}) = \nabla_\theta \mathcal{L}_{\text{supervised}}(f_{\theta_t}) + \gamma \nabla_\theta \mathcal{L}_{\text{consistency}}(f_{\theta_t}).$$

The update rule now becomes

$$
\begin{aligned}
\bar{f}_{\theta_{t+1}}^{coupled} &= \bar{f}_{\theta_t}^{coupled} - \eta \nabla_\theta \mathcal{L}(\bar{f}_{\theta_t}^{coupled})^T \phi(x) \\
&= \bar{f}_{\theta_t}^{decoupled} - \eta \nabla_\theta \mathcal{L}(\bar{f}_{\theta_t}^{decoupled})^T \phi(x) \\
&= \bar{f}_{\theta_t}^{decoupled} - \eta \big(\nabla_\theta \mathcal{L}_{\text{supervised}}(\bar{f}_{\theta_t}^{decoupled}) + \gamma \nabla_\theta \underbrace{\mathcal{L}_{\text{consistency}}(\bar{f}_{\theta_t}^{decoupled})}_{=0,\ \text{using Equation 7}}\big)^T \phi(x) \\
&= \bar{f}_{\theta_t}^{decoupled} - \eta (\nabla_\theta \mathcal{L}_{\text{supervised}}(\bar{f}_{\theta_t}^{decoupled}))^T \phi(x) \\
&= \bar{f}_{\theta_{t+1}}^{decoupled}.
\end{aligned}
$$

We have now proved for $t+1$ that $\bar{f}_{\theta_{t+1}}^{coupled} = \bar{f}_{\theta_{t+1}}^{decoupled}$, so by induction, we see the two sequences $\{\bar{f}_t^{decoupled}\}_{t=1}^n$ and $\{\bar{f}_t^{coupled}\}_{t=1}^n$ and must converge (or diverge) identically. $\square$

Lemma A.2 shows that the coupled linear ensemble converges to the decoupled ensemble. This means that the individual models tend to converge to the ensemble prediction, acting as purely ensemble distilling. This also means that if we interpret the individual models in an ensemble to be sampled from a prior, then the individual models will converge to predicting the posterior mean under training i.e. $\lim_{M \to \infty} \frac{1}{M} \sum_{m=1}^{M} f_{\theta^m}(x)$. Note that it is not always the case that individual models converge to the coupled ensemble prediction. If the coupling weight is too high, the individual models can diverge in an analogous way as too high learning rate under supervised losses in gradient descent.

## A.2. Non-linear Case

**Theorem A.3.** *Assuming all ensemble members, $f_{\theta_m}$, are non-linear models and the prediction $\bar{f}(x)$ of the ensemble is the minimizer of the sum of individual consensus losses, then $\bar{f}(x)$ does **NOT** necessarily stay constant during consensus updates.*

*Proof.* Follows directly from the derivation below, we leave it to the reader to find a function where the Taylor-approximation is exact and the second order term is non-zero. $\square$

In the non-linear case the higher-order interactions change the learning dynamics. Instead of pure knowledge distillation in the linear case, the members learn to have lower variance in the predictions. Much like in a tug-of-war where the heaviest team wins, the members with the most robust predictions will generally dominate.

To see this, consider two models $f_{\theta_1}, f_{\theta_2}$. The change in the prediction of the ensemble prediction is

$$\bar{f}_{\theta_{t+1}} - \bar{f}_{\theta_t} = \frac{1}{2}\left(f_{\theta_{t+1}^1}(x) - f_{\theta_t^1}(x)\right) + \frac{1}{2}\left(f_{\theta_{t+1}^2}(x) - f_{\theta_t^2}(x)\right). \tag{8}$$

Here $\theta_1^t$ is the parameters of model 1 after $t$ gradient updates. To calculate the ensemble prediction difference, we can approximate the parameter update $f_{\theta_{t+1}^1}(x)$ with a Taylor expansion:

$$f_{\theta_{t+1}^1}(x) \approx f_{\theta_t^1}(x) + \delta_{\theta_t^1}^T \nabla_{\theta^1} f_{\theta_t^1}(x) + \frac{1}{2}\delta_{\theta_t^1}^T \nabla_{\theta_t^1}^2 f_{\theta_t^1}(x)\delta_{\theta_t^1} + \mathcal{O}(\nabla^3 f_{\theta_t^1}(x)).$$

Calculating the change in prediction for $f_{\theta^1}$, we get

$$f_{\theta_{t+1}^1}(x) - f_{\theta_t^1}(x) \approx f_{\theta_t^1}(x) + \delta_{\theta_t^1}^T \nabla_{\theta^1} f_{\theta_t^1}(x) + \frac{1}{2}\delta_{\theta_t^1}^T \nabla_{\theta_t^1}^2 f_{\theta_t^1}(x)\delta_{\theta_t^1} - f_{\theta_t^1}(x)$$

$$= \delta_{\theta_t^1}^T \nabla_{\theta^1} f_{\theta_t^1}(x) + \frac{1}{2}\delta_{\theta_t^1}^T \nabla_{\theta_t^1}^2 f_{\theta_t^1}(x)\delta_{\theta_t^1}.$$

Here the parameter change $\delta_{\theta_t^1} = \theta_{t+1}^1 - \theta_t^1$ is given as

$$\delta_{\theta_t^1} = \theta_{t+1}^1 - \theta_t^1$$
$$= \left(\theta_t^1 - \eta\nabla_{\theta^1}\mathcal{L}(f_{\theta_t^1}(x) - \bar{f}(x))\right) - \theta_t^1$$
$$= -\eta\nabla_{\theta^1}\mathcal{L}\left(f_{\theta_t^1}(x) - \frac{1}{2}\left(f_{\theta_t^1}(x) + f_{\theta_t^2}(x)\right)\right)$$
$$= -\eta\nabla_{\theta^1}\mathcal{L}\left(\frac{f_{\theta_t^1}(x) - f_{\theta_t^2}(x)}{2}\right)$$
$$= -\frac{\eta}{2}\left(\nabla_{\theta^1}f_{\theta^1}(x)\right)\nabla\mathcal{L}\left(\frac{f_{\theta^1}(x) - f_{\theta^2}(x)}{2}\right)$$
$$= -\frac{\eta C}{2}\left(\nabla_{\theta^1}f_{\theta_t^1}(x)\right), \quad \text{where } C = \nabla\mathcal{L}\left(\frac{f_{\theta^1}(x) - f_{\theta^2}(x)}{2}\right).$$

Now

$$f_{\theta_{t+1}^1}(x) - f_{\theta_t^1}(x)$$
$$\approx \delta_{\theta_t^1}^T \nabla_{\theta^1} f_{\theta_t^1}(x) + \frac{1}{2}\delta_{\theta_t^1}^T \nabla_{\theta_t^1}^2 f_{\theta_t^1}(x)\delta_{\theta_t^1}$$
$$= -\frac{\eta C}{2}\left(\nabla_{\theta^1}f_{\theta_t^1}(x)^T\nabla_{\theta^1}f_{\theta_t^1}(x)\right) + \frac{1}{2}\delta_{\theta_t^1}^T \nabla_{\theta_t^1}^2 f_{\theta_t^1}(x)\delta_{\theta_t^1}$$
$$= -\frac{\eta C}{2}\|\nabla_{\theta^1}f_{\theta_t^1}(x)\|_2^2 + \frac{\eta^2 C^2}{8}\left(\nabla_{\theta^1}f_{\theta_t^1}(x)\right)^T\nabla_{\theta_t^1}^2 f_{\theta_t^1}(x)\left(\nabla_{\theta^1}f_{\theta_t^1}(x)\right)$$
$$= -\frac{\eta C}{2}\left(\|\nabla_{\theta^1}f_{\theta_t^1}(x)\|_2^2 - \frac{1}{4}\eta C\left(\nabla_{\theta^1}f_{\theta_t^1}(x)\right)^T\nabla_{\theta_t^1}^2 f_{\theta_t^1}(x)\left(\nabla_{\theta^1}f_{\theta_t^1}(x)\right)\right).$$

Similar arguments can be made for model $f_{\theta^2}$, by assuming that $\mathcal{L}$ is symmetric, such that $\nabla\mathcal{L}(-x) = -C$. Now we get

$$f_{\theta^2_{t+1}}(x) - f_{\theta^2_t}(x) \approx$$
$$\frac{\eta C}{2}\left(||\nabla_{\theta^2}f_{\theta^2_t}(x)||_2^2 + \frac{1}{4}\eta C\left(\nabla_{\theta^2}f_{\theta^2_t}(x)\right)^T\nabla^2_{\theta^2}f_{\theta^2_t}(x)\left(\nabla_{\theta^2}f_{\theta^2_t}(x)\right)\right).$$

Combining this, we can calculate the change in ensemble prediction to be

$$\bar{f}_{\theta_{t+1}} - \bar{f}_{\theta_t} = \frac{\eta C}{4}\left(\underbrace{||\nabla_{\theta^2}f_{\theta^2_t}(x)||_2^2 - ||\nabla_{\theta^1}f_{\theta^1_t}(x)||_2^2}_{\text{First-order difference in gradient}}\right.$$
$$+\frac{1}{4}\eta C\underbrace{\left(\left(\nabla_{\theta^1}f_{\theta^1_t}(x)\right)^T\nabla^2_{\theta^1}f_{\theta^1_t}(x)\left(\nabla_{\theta^1}f_{\theta^1_t}(x)\right)\right) + \left(\left(\nabla_{\theta^2}f_{\theta^2_t}(x)\right)^T\nabla^2_{\theta^2}f_{\theta^2_t}(x)\left(\nabla_{\theta^2}f_{\theta^2_t}(x)\right)\right)}_{\text{Second-order curvature}}$$
$$+ \mathcal{O}(\nabla^3 f_{\theta_t}(x))\bigg).$$

To break down the result of the above derivation consider the example where $f_{\theta^1}$ has a more robust prediction i.e. has a lower squared gradient $f_{\theta^2}$ and both models have positive curvature. In addition say that $f_{\theta^1_t}(x) > f_{\theta^2_t}(x)$, so we get the loss gradient point towards $f_{\theta^1_t}(x)$ i.e. $C > 0$. In this case the first order difference in gradients must be positive, and the ensemble prediction moves more towards $f_{\theta^1}(x)$.

While we do see that increasing the curvature of $f_{\theta^1}(x)$ increases the ensemble prediction, but this effect is scaled by the gradient. This means that, in addition to it be a second order effect, it is also dampened by the gradient size. This means that if an ensemble member has robust predictions, i.e. large parameter spaces with low gradient size and therefore curvature, it will serve as attractors for the ensemble consensus.

### A.3. Experimental Validation

To investigate the effect of the ensemble consensus inducing more robust models we log the gradient norm and trace of the Hessian for different coupling strengths. We estimate the trace of the Hessian using Hutchinson's method (Hutchinson, 1989). Both quantities are estimated with 32000 (1000 batches). The results are shown in Figure 4 and the model MAE in Figure 5.

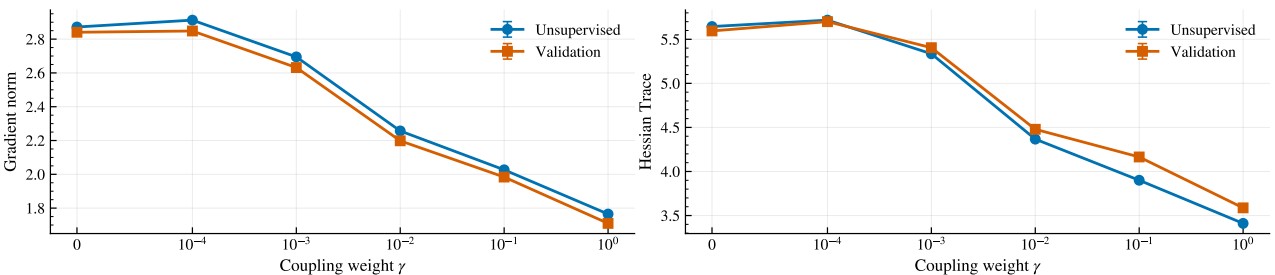

*Figure 4.* The gradient norm (left) and Hessian of the prediction (right) of the models of different data for different coupling strengths. Experimental setup follows the PaiNN setup section 6.1 experiments on target ZPVE.

From the experiments we see that the gradient norm and Hessian trace decreases with the coupling strength as predicted by the theory. This shows the model's predictions become more robust with the coupling strength, serving as a kind of regularisation. We hypothesise this to be the main reason for the models trained with ensemble consensus can improve over the corresponding deep ensemble. The ensemble consensus converges towards a more stable model instead of converging to a posterior mean. The regularisation from coupling can be too large, as seen in figure 5, where the optimal value is $10^{-3}$. Interestingly, as the models are trained with consensus loss on the unsupervised data, we would expect the gradient norm and Hessian trace to be lower on unsupervised data than the unseen validation data. Instead they are roughly similar, suggesting the effect generalises to unseen data. This is further backed by the prediction error (MAE) being virtually identical for the unsupervised and validation data.

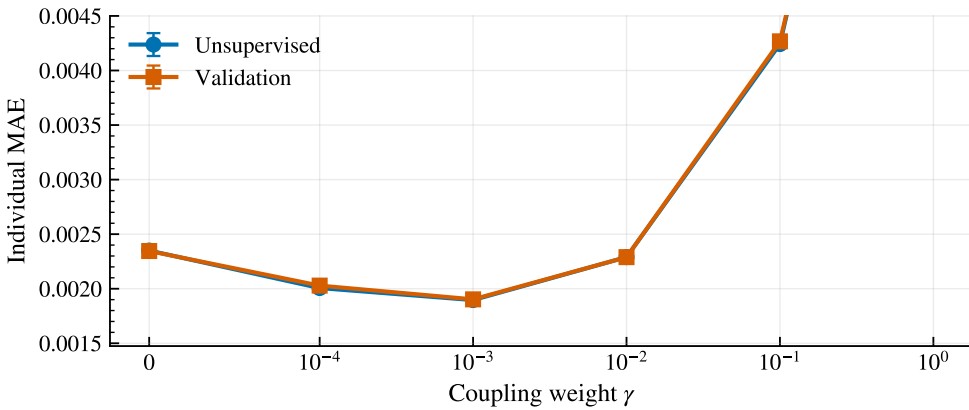

*Figure 5.* The MAE of the models for different coupling strengths. Experimental setup follows the PaiNN setup section 6.1 experiments on target ZPVE.

# B. Extended Studies

## B.1. Scaling with Number of Ensemble Members

We also investigate the predictive accuracy scaling with the number of ensemble members to larger than 4 sizes. Ensembles of these sizes were not feasible to do on any of the graph datasets, so we instead use the original computer vision version of CIFAR-10. This also validates that our method works for other domains than graphs. We use ResNet-18 (He et al., 2016) with 5,000 labeled and 40,000 unlabeled data-points without any data augmentations. We performed an exhaustive hyperparameter sweep using a single seed over learning rate (0.1, 0.075, 0.05, 0.025, 0.01, 0.0075, 0.005, 0.0025, 0.001), and weight decay (0.01, 0.025, 0.05, 0.075, 0.1, 0.25, 0.5) for the purely supervised model. The number of epochs and learning rate annealing was fixed at a number informally found to work. The parameters of best performing model on validation accuracy at the last epoch was selected. The optimal values can be found in 17. The coupling weight was fixed kept at $\gamma = 1$.

The hyper-parameters can be found in Appendix C.3. From the accuracy results in Table 5 and calibration scores in section B.2, we see a significant increase in accuracy and calibration scores going from a single model to a coupled ensemble with just two models. Interestingly, the individual prediction accuracy of a model trained in a coupled ensemble of two models outperforms the ensemble prediction from all decoupled ensemble sizes tested. This highlights the semi-supervised effect from using unlabeled data for training. Looking at the calibration metrics in Appendix B.2, we see that the calibration results for the coupled ensemble are worse than the uncoupled one. This is often seen in self-supervised learning, as the "self-validating" training can result in worse calibration from confirmation bias (Arazo et al., 2020; Mishra et al., 2024). Surprisingly, we see the individual calibration improving over the decoupled model (i.e., a single model), and also improving as the number of ensemble members increases.

*Table 5.* Predictive accuracy (%) on CIFAR-10 validation, comparing Decoupled and Coupled models. The values represent mean $\pm$ 1.96 standard error of the mean.

|  | Individual Accuracy % | | Ensemble Accuracy (%) | |
| --- | --- | --- | --- | --- |
| Ensemble size | Decoupled | Coupled | Decoupled | Coupled |
| 1 | $59.08_{\pm 1.35}$ | $\cdots$ | $\cdots$ | $\cdots$ |
| 2 | $\vdots$ | $66.36_{\pm 0.45}$ | $62.51_{\pm 0.40}$ | $66.96_{\pm 0.47}$ |
| 4 | $\vdots$ | $67.24_{\pm 0.40}$ | $64.65_{\pm 0.46}$ | $67.92_{\pm 0.49}$ |
| 8 | $\vdots$ | $67.64_{\pm 0.35}$ | $65.73_{\pm 0.51}$ | $68.34_{\pm 0.34}$ |
| 16 | $\vdots$ | $67.75_{\pm 0.32}$ | $66.41_{\pm 0.57}$ | $68.54_{\pm 0.45}$ |
| 32 | $\vdots$ | $67.75_{\pm 0.30}$ | $66.64_{\pm 0.37}$ | $68.52_{\pm 0.35}$ |

*Table 6.* GCN performance (MAE) for different ensemble sizes on peptides-struct. Comparison between Supervised and Supervised + SSL training. Results are reported as mean $\pm$ standard error.

| Size (M) | Individual member | | Ensemble | |
|---|---|---|---|---|
| | Supervised | Supervised + SSL | Supervised | Supervised + SSL |
| 1 | $0.3005_{\pm 0.0064}$ | $\ldots$ | $\ldots$ | $\ldots$ |
| 2 | $\vdots$ | $0.2858_{\pm 0.0057}$ | $0.2958_{\pm 0.0036}$ | $0.2852_{\pm 0.0057}$ |
| 3 | $\vdots$ | $0.2864_{\pm 0.0057}$ | $0.2946_{\pm 0.0040}$ | $0.2860_{\pm 0.0054}$ |
| 4 | $\vdots$ | $0.2870_{\pm 0.0037}$ | $0.2926_{\pm 0.0057}$ | $0.2867_{\pm 0.0039}$ |
| 6 | $\vdots$ | $0.2856_{\pm 0.0060}$ | $0.2921_{\pm 0.0057}$ | $0.2852_{\pm 0.0056}$ |
| 8 | $\vdots$ | $0.2874_{\pm 0.0045}$ | $0.2916_{\pm 0.0057}$ | $0.2871_{\pm 0.0042}$ |
| 16 | $\vdots$ | $0.2870_{\pm 0.0052}$ | $0.2920_{\pm 0.0052}$ | $0.2869_{\pm 0.0053}$ |

## B.2. Calibration Metrics on CIFAR-10

*Table 7.* NLL on CIFAR-10, comparing decoupled and coupled models. The values represent mean $\pm$ 1.96 standard error of the mean.

| | NLL $\downarrow$ | | | |
|---|---|---|---|---|
| | Individual member | | Ensemble | |
| Ensemble size | Decoupled | Coupled | Decoupled | Coupled |
| 1 | $1.543_{\pm 0.109}$ | $\ldots$ | $\ldots$ | $\ldots$ |
| 2 | $\vdots$ | $1.217_{\pm 0.021}$ | $1.267_{\pm 0.020}$ | $1.161_{\pm 0.021}$ |
| 4 | $\vdots$ | $1.169_{\pm 0.019}$ | $1.121_{\pm 0.012}$ | $1.096_{\pm 0.019}$ |
| 8 | $\vdots$ | $1.142_{\pm 0.017}$ | $1.048_{\pm 0.015}$ | $1.064_{\pm 0.016}$ |
| 16 | $\vdots$ | $1.126_{\pm 0.015}$ | $1.007_{\pm 0.015}$ | $1.047_{\pm 0.014}$ |
| 32 | $\vdots$ | $1.123_{\pm 0.019}$ | $0.990_{\pm 0.011}$ | $1.042_{\pm 0.018}$ |

*Table 8.* AUC-ROC on CIFAR-10, comparing decoupled and coupled models. The values represent mean $\pm$ 1.96 standard error of the mean.

| | AUC-ROC $\uparrow$ | | | |
|---|---|---|---|---|
| | Individual member | | Ensemble | |
| Ensemble size | Decoupled | Coupled | Decoupled | Coupled |
| 1 | $.8885_{\pm .0075}$ | $\ldots$ | $\ldots$ | $\ldots$ |
| 2 | $\vdots$ | $.9250_{\pm .0020}$ | $.9125_{\pm .0021}$ | $.9292_{\pm .0020}$ |
| 4 | $\vdots$ | $.9295_{\pm .0019}$ | $.9266_{\pm .0016}$ | $.9349_{\pm .0019}$ |
| 8 | $\vdots$ | $.9316_{\pm .0019}$ | $.9336_{\pm .0021}$ | $.9377_{\pm .0018}$ |
| 16 | $\vdots$ | $.9323_{\pm .0016}$ | $.9384_{\pm .0019}$ | $.9386_{\pm .0015}$ |
| 32 | $\vdots$ | $.9329_{\pm .0019}$ | $.9409_{\pm .0017}$ | $.9394_{\pm .0019}$ |

*Table 9.* ECE on CIFAR-10, comparing decoupled and coupled models. The values represent mean $\pm$ 1.96 standard error of the mean.

| | ECE $\downarrow$ | | | |
| | Individual member | | Ensemble | |
| Ensemble size | Decoupled | Coupled | Decoupled | Coupled |
|---|---|---|---|---|
| 1 | $.2210_{\pm.0357}$ | $\cdots$ | $\cdots$ | $\cdots$ |
| 2 | $\vdots$ | $.1713_{\pm.0041}$ | $.1128_{\pm.0057}$ | $.1512_{\pm.0049}$ |
| 4 | $\vdots$ | $.1609_{\pm.0034}$ | $.0591_{\pm.0043}$ | $.1369_{\pm.0040}$ |
| 8 | $\vdots$ | $.1548_{\pm.0031}$ | $.0320_{\pm.0043}$ | $.1301_{\pm.0035}$ |
| 16 | $\vdots$ | $.1494_{\pm.0028}$ | $.0243_{\pm.0033}$ | $.1235_{\pm.0030}$ |
| 32 | $\vdots$ | $.1485_{\pm.0044}$ | $.0207_{\pm.0043}$ | $.1226_{\pm.0043}$ |

*Table 10.* Brier score on CIFAR-10, comparing decoupled and coupled models. The values represent mean $\pm$ 1.96 standard error of the mean.

| | Brier $\downarrow$ | | | |
| | Individual member | | Ensemble | |
| Ensemble size | Decoupled | Coupled | Decoupled | Coupled |
|---|---|---|---|---|
| 1 | $.4854_{\pm.0271}$ | $\cdots$ | $\cdots$ | $\cdots$ |
| 2 | $\vdots$ | $.5594_{\pm.0042}$ | $.4585_{\pm.0041}$ | $.5530_{\pm.0040}$ |
| 4 | $\vdots$ | $.5654_{\pm.0040}$ | $.4422_{\pm.0047}$ | $.5572_{\pm.0040}$ |
| 8 | $\vdots$ | $.5676_{\pm.0048}$ | $.4316_{\pm.0050}$ | $.5588_{\pm.0047}$ |
| 16 | $\vdots$ | $.5652_{\pm.0044}$ | $.4283_{\pm.0049}$ | $.5563_{\pm.0043}$ |
| 32 | $\vdots$ | $.5649_{\pm.0030}$ | $.4263_{\pm.0033}$ | $.5558_{\pm.0030}$ |

## B.3. Non-Chemical GNN+ datasets

Results for non-chemical GNN+ datasets are shown in Table 11. Note the consensus and mean-teacher run for the GatedGCN models were not computed, as the models were too large to fit in memory.

*Table 11.* Performance on non-molecule-related benchmarks, comparing supervised models with those using additional self-supervised learning (SSL). Results are shown for individual models (Individual) and the full ensemble (Ensemble). Results are the mean $\pm 1.96$ standard error of the mean over 5 different seeds. * denotes models that did not fit in our 32GB video-memory.

| Dataset | Training | Metric | GCN | | GIN | | GatedGCN | |
|---|---|---|---|---|---|---|---|---|
| | | | Individual | Ensemble | Individual | Ensemble | Individual | Ensemble |
| CIFAR-10 | Supervised | Acc (%)↑ | $50.44_{\pm 0.33}$ | $55.38_{\pm 0.49}$ | $50.46_{\pm 0.34}$ | $53.90_{\pm 0.50}$ | $57.69_{\pm 0.34}$ | $61.23_{\pm 0.45}$ |
| | Consensus | | $55.33_{\pm 0.31}$ | $57.11_{\pm 0.42}$ | $54.30_{\pm 0.36}$ | $55.60_{\pm 0.31}$ | * | |
| | Mean teacher | | $50.64_{\pm 0.28}$ | | $50.99_{\pm 0.86}$ | - | $57.80_{\pm 0.57}$ | - |
| MNIST | Supervised | Acc (%)↑ | $96.61_{\pm 0.07}$ | $96.97_{\pm 0.04}$ | $96.26_{\pm 0.10}$ | $96.73_{\pm 0.13}$ | $96.96_{\pm 0.05}$ | $97.38_{\pm 0.11}$ |
| | Consensus | | $96.82_{\pm 0.08}$ | $96.93_{\pm 0.11}$ | $96.68_{\pm 0.09}$ | $96.82_{\pm 0.11}$ | $97.48_{\pm 0.06}$ | $97.57_{\pm 0.07}$ |
| | Mean teacher | | $96.55_{\pm 0.06}$ | - | $96.31_{\pm 0.11}$ | - | $96.84_{\pm 0.13}$ | - |
| CLUSTER | Supervised | AP ↑ | $69.72_{\pm 0.98}$ | $72.17_{\pm 0.12}$ | $69.72_{\pm 0.98}$ | $72.17_{\pm 0.12}$ | $69.72_{\pm 0.98}$ | $76.84_{\pm 0.09}$ |
| | Consensus | | $71.61_{\pm 0.31}$ | $72.44_{\pm 0.19}$ | $71.61_{\pm 0.31}$ | $72.44_{\pm 0.19}$ | * | |
| | Mean teacher | | $69.53_{\pm 4.52}$ | - | $69.10_{\pm 1.50}$ | - | $72.07_{\pm 7.91}$ | - |

### B.4. Different label amounts

We also include an investigation into how the ensemble coupling is influenced by different number of labeled versus unlabeled data. Specifically we vary the amount of training data that is labeled of the entire pool. The results on QM9 target ZPVE (target 7) can be seen in Figure 6. We observe a smooth improvement across all fractions without any collapse at low label fractions. Interestingly, at $50\%$ of labeled data, the individual models in coupled ensemble obtains the same error as the an individual model trained on the full data.

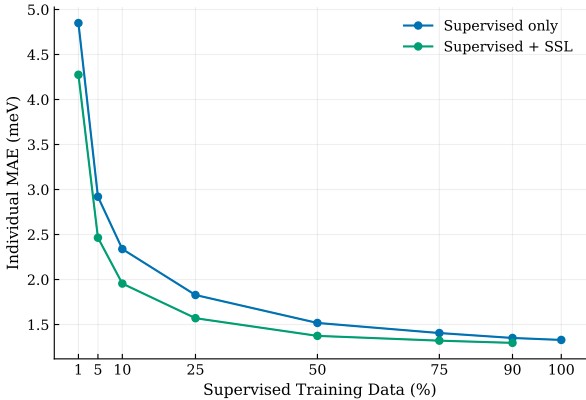

*Figure 6.* Validation MAE as a function of the amount of data labeled for QM9 target ZPVE (target 7). The results are averaged over 5 seeds.

### B.5. Variance Inside the Ensemble

We investigate the impact the coupling strength has on the diversity of the ensemble. Increasing the coupling strength should directly reduce the diversity, especially on the unlabeled data, which is also seen in Figure 3 (left) and 7. For completion, we also plot the variance within the ensemble over epochs in Figure 8.

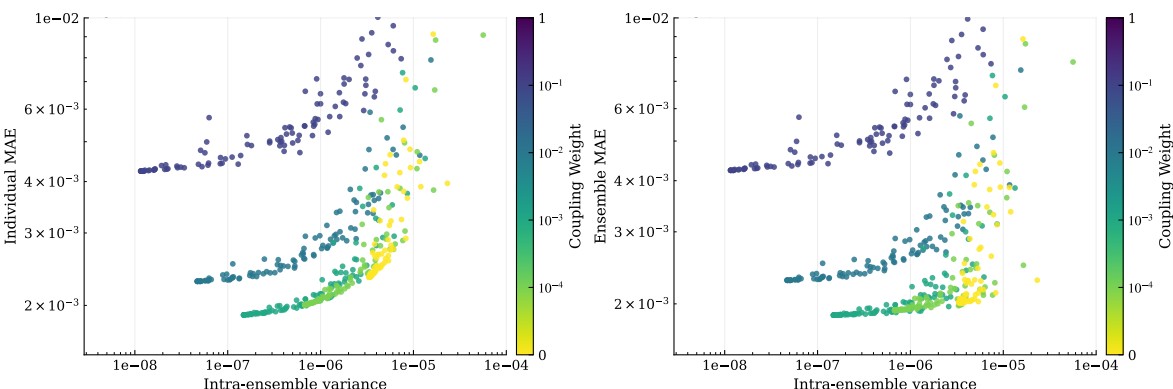

*Figure 7.* Error of the individual models (left) and ensemble (right) as a function of the variance within the ensemble across a training run. The data points are coloured based on the coupling strength on QM9 target 7.

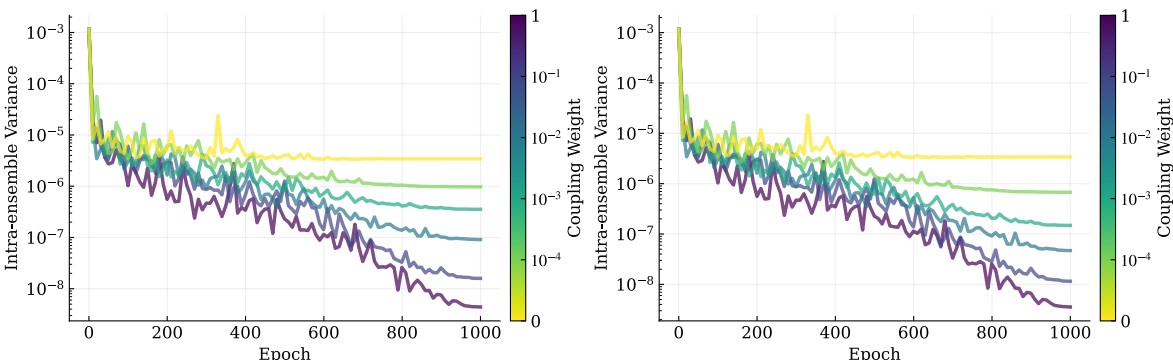

*Figure 8.* Variance within the ensemble over epochs for different coupling weights for validation data (left) and unsupervised data (right). The data points are coloured based on the coupling strength on QM9 target 7.

## B.6. Distribution Scaffolding

We also investigate the behavior of ensemble consensus when the unlabeled data is from a different distribution. Specifically, we use scaffold splitting with Butina-clustering (Butina, 1999) for QM9. Specifically, we split QM9 into two distributions for the unsupervised data and one for the rest. We use the same hyperparameters as QM9. The results can be seen in Table 12. For all targets we see an improvement from ensemble consensus, with an individual consensus model outperforming a uncoupled ensemble for all targets except $\langle R^2 \rangle$ and G. This suggests ensemble coupling is effective even when the unlabeled distribution is not too far away from the labeled distribution. Note, that the non-consensus runs for target 0 perform poorly, presumably due to the distributional shift being large enough to destabilise the runs.

*Table 12.* PaiNN performance (MAE) on QM9 targets with Butina Scaffolding. Results are reported as mean $\pm 1.96$ standard error of the mean over 5 seeds.

| Target | Unit | Individual Member | | Ensemble (M=4) | |
|---|---|---|---|---|---|
| | | Supervised | Supervised + SSL | Supervised | Supervised + SSL |
| $\mu$ | D | $.6168_{\pm.1741}$ | $.0593_{\pm.0038}$ | $.6106_{\pm.1740}$ | $.05861_{\pm.0038}$ |
| $\alpha$ | $a_0^3$ | $.1496_{\pm.0029}$ | $.1244_{\pm.0026}$ | $.1315_{\pm.0021}$ | $.1217_{\pm.0026}$ |
| $\epsilon_{\text{HOMO}}$ | meV | $79.635_{\pm1.294}$ | $74.641_{\pm1.549}$ | $75.807_{\pm1.416}$ | $73.612_{\pm1.537}$ |
| $\epsilon_{\text{LUMO}}$ | meV | $61.325_{\pm.563}$ | $57.290_{\pm.422}$ | $58.771_{\pm.650}$ | $56.732_{\pm.431}$ |
| $\Delta\epsilon$ | meV | $126.920_{\pm3.936}$ | $118.369_{\pm2.575}$ | $121.950_{\pm3.906}$ | $116.746_{\pm2.229}$ |
| $\langle R^2 \rangle$ | $a_0^2$ | $.8634_{\pm.0406}$ | $.8337_{\pm.0197}$ | $.6675_{\pm.0378}$ | $.6499_{\pm.0201}$ |
| ZPVE | $a_0^2$ | $2.327_{\pm.023}$ | $1.960_{\pm.026}$ | $2.018_{\pm.022}$ | $1.902_{\pm.027}$ |
| $U_0$ | meV | $23.813_{\pm.557}$ | $19.524_{\pm.321}$ | $20.039_{\pm.499}$ | $18.678_{\pm.300}$ |
| $U$ | meV | $23.994_{\pm.382}$ | $19.787_{\pm.460}$ | $20.160_{\pm.366}$ | $18.937_{\pm.435}$ |
| $H$ | meV | $24.021_{\pm.476}$ | $19.777_{\pm.518}$ | $20.191_{\pm.457}$ | $18.934_{\pm.528}$ |
| $G$ | meV | $24.475_{\pm0.527}$ | $23.157_{\pm1.833}$ | $20.748_{\pm.459}$ | $22.656_{\pm1.987}$ |
| $C_v$ | $\frac{\text{cal}}{\text{mol K}}$ | $.0558_{\pm.0005}$ | $.0464_{\pm.0006}$ | $.0484_{\pm.0003}$ | $.0449_{\pm.0007}$ |

## C. Hyperparameters

### C.1. QM9

Our hyperparameter search for QM9 followed a two-step process. First, we started with baseline hyperparameters from a fully supervised setting and tuned the learning rate and weight decay for a single model on the 10% labeled data subset. Second, using these optimized parameters, we then tuned the coupling weight ($\gamma$) for the size-4 ensemble by searching over $\{1.0, 0.1, 0.01, 0.001, 0.0001\}$. The coupling weight swept for the mean-teacher was $\{0.9, 0.95, 0.99, 0.995, 0.999\}$. Final a architectural and training configurations are detailed in Table 13 and Table 14.

#### C.1.1. PSEUD$\sigma$ BASELINE IMPLEMENTATION

We re-implemented the Uncertainty-Aware Pseudo-labeling (PSEUD$\sigma$) method (Huang et al., 2022) for the PaiNN architecture, as the original study did not evaluate this model. To ensure a direct comparison, we utilized the identical 10% labeled / 90% unlabeled data split. Our implementation features a PaiNN backbone equipped with an evidential head to output the required prior parameters ($\gamma, v, \alpha, \beta$). We trained using AdamW (1e-4 LR, 1e-4 WD) with a batch size of 32. The training schedule consisted of a 1000-epoch initial training phase on the labeled data, followed by 15 outer-loop episodes ($M$) of 100 inner-loop epochs ($K$) each. We adopted the original paper's recommended low-data hyperparameters, including an evidential regularization coefficient ($\lambda$) of 0.5 and epistemic uncertainty for adaptive weighting. Consistent with the PSEUD$\sigma$ strategy, our cosine annealing learning rate scheduler was re-initialized at the start of each of the 15 episodes.

*Table 13.* Hyperparameter Configuration for QM9. These are fixed across all targets.

| Hyperparameter | Value |
|---|---|
| **Training** | |
| Batch size | 32 |
| Epochs | 1000 |
| Optimizer | AdamW |
| Scheduler | Cosine annealing |
| **Coupling** | |
| Unsupervised loss criterion | L2 |

*Table 14.* Additional hyperparameter Configuration for QM9 for different targets.

| Target | Learning rate | Weight decay | Coupling weight | Mean teacher decay |
|---|---|---|---|---|
| $\mu$ | 1e-3 | 1e-3 | 0.1 | 0.995 |
| $\alpha$ | 1e-4 | 1e-3 | 0.1 | 0.99 |
| $\epsilon_{\text{HOMO}}$ | 1e-3 | 0 | 0.01 | 0.95 |
| $\epsilon_{\text{LUMO}}$ | 5e-4 | 1e-6 | 0.01 | 0.9 |
| $\Delta\epsilon$ | 1e-3 | 0 | 0.01 | 0.99 |
| $\langle R^2 \rangle$ | 5e-4 | 1e-4 | 0.1 | 0.99 |
| ZPVE | 5e-4 | 1e-5 | 0.001 | 0.99 |
| $U_0$ | 1e-4 | 1e-4 | 0.01 | 0.99 |
| $U$ | 1e-4 | 0 | 0.01 | 0.9 |
| $H$ | 1e-4 | 1e-4 | 0.01 | 0.9 |
| $G$ | 1e-4 | 1e-5 | 0.01 | 0.995 |
| $C_v$ | 1e-4 | 1e-5 | 0.01 | 0.995 |

## C.2. GNN+ Datasets

We keep the hyperparameters for the different datasets and models the same as in the original paper, except for the number of epochs, weight decay, and learning rate. As we are training with $10\%$ of the original data, we double the number of epochs to mitigate the fewer parameter updates. We then made a two-step hyper-parameter sweep; initially the learning rate using original weight decay values, and afterwards the weight decay using the found best learning rates. The learning rates investigated were $(0.25, 0.5, 1.0, 2.0, 4.0)$ times the original learning rate value for that model and dataset. The weight decays investigated was $(10^{-6}, 10^{-5}, 10^{-4}, 10^{-3}, 10^{-2}, 10^{-1}, 0)$. We could not simply multiply the weight decay values by a fixed factor, as some of the original weight decay values were 0. These sweeps were performed for a single uncoupled model following the same tuning procedure as in the original paper. Notably, this means that the predictive accuracy report from each run is the best validation performance seen during any of the epochs. The found learning rates are listed in Table 15, and weight decays Table 16 below. The train, validation, and test splits follow the same procedure as (Luo et al., 2025). Each seed shuffles the labeled and unlabeled part of the training data.

The SSL parameters were selected based on the best performing values on the validation score on ZINC. The mean-teacher values investigated was $(0.9, 0.99, 0.995, 0.999)$, and the coupling weight for the consensus and pair-wise methods were $(0.25, 0.5, 0.75, 1, 1.25, 1.5, 1.75, 2.0)$. The optimal value of mean-teacher was found to be 0.999, and coupling weight for the consensus learning was 1.0, and the pairwise loss was tied between 0.5 and 0.75, so we went with 0.5 based on the recommendations in (Filipiak et al., 2021).

*Table 15.* Tuned learning rates for GNN models across datasets.

| Dataset | GCN | GIN | GatedGCN |
|---|---|---|---|
| CIFAR-10 | 0.002 | 0.0005 | 0.001 |
| CLUSTER | 0.0005 | 0.0005 | 0.002 |
| ogbg-molhiv | 0.0001 | 0.00005 | 0.0004 |
| MNIST | 0.001 | 0.002 | 0.001 |
| ogbg-molpcba | 0.000125 | 0.000125 | 0.00025 |
| peptides-func | 0.0005 | 0.002 | 0.002 |
| peptides-struct | 0.002 | 0.0005 | 0.002 |
| ogbg-ppa | 0.0006 | 0.0012 | 0.0003 |
| ZINC | 0.004 | 0.001 | 0.004 |

*Table 16.* Tuned weight decays for GNN models across datasets.

| Dataset | GCN | GIN | GatedGCN |
|---|---|---|---|
| CIFAR-10 | $10^{-2}$ | $10^{-1}$ | $10^{-2}$ |
| CLUSTER | 0 | $10^{-1}$ | $10^{-6}$ |
| ogbg-molhiv | $10^{-3}$ | $10^{-1}$ | $10^{-5}$ |
| MNIST | $10^{-1}$ | $10^{-2}$ | $10^{-5}$ |
| ogbg-molpcba | $10^{-1}$ | $10^{-2}$ | $10^{-5}$ |
| peptides-func | 0 | $10^{-1}$ | $10^{-3}$ |
| peptides-struct | $10^{-3}$ | $10^{-5}$ | $10^{-1}$ |
| ogbg-ppa | $10^{-1}$ | $10^{-1}$ | $10^{-2}$ |
| ZINC | $10^{-1}$ | $10^{-5}$ | $10^{-3}$ |

### C.3. CIFAR-10

The hyperparameter configurations for CIFAR-10 are shown in Table 17.

*Table 17.* Hyperparameter Configuration for CIFAR-10.

| Hyperparameter | Value |
|---|---|
| **Learning Rate** | |
| Learning rate | 0.005 |
| Annealing method | Step |
| Step size | 1 |
| Learning rate reduction | 0.975 |
| **Regularization** | |
| L2 Weight Decay | 0.075 |
| **Optimizer** | |
| Optimizer | SGD |
| Momentum | 0.9 |
| **Training** | |
| Epochs | 250 |
| **Loss Function** | |
| Coupled loss weighting | 1.0 |
| Ensemble coupled loss | KL-divergence |
| Supervised loss | Cross-entropy |

## D. Calibration Scores for the ogbg-molhiv

We also investigate the calibration on the ogbg-molhiv benchmark. We do not investigate the datasets ogbg-pcba and peptides functional due to the to the large skewing of classes and missing values. The results are included in Table 18 and Table 19. We see across different architectures that the coupling of the ensemble improves the calibration scores, especially NLL. One notable exception is the MCE score for the GIN ensemble model, where the coupled ensemble becomes significantly worse.

*Table 18.* Individual Performance on the ogbg-molhiv dataset

| Metric | GCN | | GIN | | GatedGCN | |
|---|---|---|---|---|---|---|
| | Decoupled | Coupled | Decoupled | Coupled | Decoupled | Coupled |
| Accuracy ↑ | $95.78_{\pm0.38}$ | $96.18_{\pm0.48}$ | $95.97_{\pm0.68}$ | $96.30_{\pm0.35}$ | $95.66_{\pm0.72}$ | $96.01_{\pm0.57}$ |
| ROC-AUC ↑ | $.721_{\pm.0193}$ | $.731_{\pm.0218}$ | $.733_{\pm.017}$ | $.734_{\pm.015}$ | $.731_{\pm.008}$ | $.736_{\pm.007}$ |
| NLL ↓ | $.375_{\pm.185}$ | $.230_{\pm.0662}$ | $.147_{\pm.015}$ | $.140_{\pm.012}$ | $.200_{\pm.033}$ | $.180_{\pm.023}$ |
| ECE ↓ | $.0312_{\pm.0092}$ | $.0246_{\pm.0039}$ | $.0113_{\pm.0045}$ | $.0105_{\pm.0048}$ | $.0232_{\pm.0069}$ | $.0201_{\pm.0049}$ |
| MCE ↓ | $.2041_{\pm.0994}$ | $.2058_{\pm.0763}$ | $.1113_{\pm.0620}$ | $.1058_{\pm.0246}$ | $.1154_{\pm.0399}$ | $.0985_{\pm.0287}$ |

*Table 19.* Ensemble Performance on the ogbg-molhiv dataset

| Metric | GCN | | GIN | | GatedGCN | |
|---|---|---|---|---|---|---|
| | Decoupled | Coupled | Decoupled | Coupled | Decoupled | Coupled |
| Accuracy ↑ | $96.66_{\pm0.33}$ | $96.60_{\pm0.20}$ | $96.11_{\pm0.66}$ | $96.39_{\pm0.33}$ | $96.03_{\pm0.62}$ | $96.12_{\pm0.56}$ |
| ROC-AUC ↑ | $.7350_{\pm.0228}$ | $.7357_{\pm.0212}$ | $.7346_{\pm.0165}$ | $.7347_{\pm.0153}$ | $.7341_{\pm.0107}$ | $.7383_{\pm.0073}$ |
| NLL ↓ | $.2437_{\pm.1051}$ | $.1760_{\pm.0275}$ | $.1432_{\pm.0130}$ | $.1383_{\pm.0108}$ | $.1821_{\pm.0249}$ | $.1729_{\pm.0208}$ |
| ECE ↓ | $.0261_{\pm.0057}$ | $.0224_{\pm.0046}$ | $.0121_{\pm.0039}$ | $.0109_{\pm.0037}$ | $.0201_{\pm.0051}$ | $.0193_{\pm.0045}$ |
| MCE ↓ | $.2587_{\pm.0793}$ | $.2617_{\pm.0564}$ | $.1585_{\pm.0760}$ | $.1933_{\pm.0852}$ | $.1566_{\pm.0576}$ | $.1533_{\pm.0251}$ |

# E. Ablation Studies

### E.1. Soft or Hard labels for Classification

Often semi-supervised methods use some form of "hard-labeling" as the consistency target. Usually, this is implemented as setting the ensemble target for an unlabeled datapoint to be the most likely label, as predicted by the individual model (Filipiak et al., 2021; Tarvainen & Valpola, 2017) or the ensemble (Platanios, 2018). This removes the underlying uncertainty information of the estimates, and risking drastically reducing the calibration of the model by making it overconfident. The motivation for using hard-labeling is the assumption of label smoothness, as it forces the model to pick the same label for data points close together. We investigate this assumption in table 20. The results on accuracy show that hard-labelling slightly benefits the accuracy, it comes at the cost of worse calibration metrics such as ECE and MCE for the individual models. One explanation for the small increase in accuracy is the label-smoothens assumption can be violated for graphs and especially molecules.

*Table 20.* Calibration metrics on graph CIFAR-10.

| Metric | Individual | | Ensemble | |
|---|---|---|---|---|
| | Mean | Hard Label | Mean | Hard Label |
| Accuracy (%)↑ | $56.0220_{\pm0.2233}$ | $56.2020_{\pm0.5595}$ | $56.7640_{\pm0.2742}$ | $57.1920_{\pm0.4124}$ |
| ROC ↑ | $.9040_{\pm.0017}$ | $.8936_{\pm.0025}$ | $.7598_{\pm.0015}$ | $.7621_{\pm.0022}$ |
| F1 ↑ | $.5586_{\pm.0021}$ | $.5607_{\pm.0051}$ | $.5661_{\pm.0023}$ | $.5706_{\pm.0034}$ |
| ECE ↓ | $.1514_{\pm.0030}$ | $.3034_{\pm.0052}$ | $.4324_{\pm.0027}$ | $.4281_{\pm.0041}$ |
| MCE ↓ | $.2307_{\pm.0030}$ | $.4252_{\pm.0141}$ | $.4324_{\pm.0027}$ | $.4281_{\pm.0041}$ |

### E.2. Pairwise or Coupled Ensemble

There is a strong theoretical connection between the pairwise loss between ensemble members used in n-CPS (Filipiak et al., 2021) and the coupled ensemble loss presented in this work. The pairwise loss is defined as

$$\mathcal{L}_{\text{pairwise}}\big(f_{\theta_i}(x)\big) = \frac{1}{M-1} \sum_{m=1}^{M} \mathcal{L}\big(f_{\theta_i}(x) - f_{\theta_m}(x)\big).$$

For a convex loss $\mathcal{L}$ that can be written on the form $\mathcal{L}(x - y)$, then Jensen's inequality yields

$$
\begin{aligned}
\mathcal{L}\big(f_{\theta_i}(x) - \mathbb{E}_m[f_{\theta_m}(x)]\big) &= \mathcal{L}\big(\mathbb{E}_m[f_{\theta_i}(x) - f_{\theta_m}(x)]\big) \\
&\leq \mathbb{E}_m[\mathcal{L}\big(f_{\theta_i}(x) - f_{\theta_m}(x)\big) \\
&= \frac{1}{M}\sum_{m=1}^{M}\mathcal{L}\big(f_{\theta_i}(x) - f_{\theta_m}(x)\big) \\
&\leq \frac{1}{M-1}\sum_{m=1}^{M}\mathcal{L}\big(f_{\theta_i}(x) - f_{\theta_m}(x)\big).
\end{aligned}
$$

As $f_{\theta_i}(x) - f_{\theta_m}(x) = 0$ if $i = m$ this upper bound is exactly the n-CPS loss. In general this upper bound is not tight, but if $M = 2$ and $\mathcal{L}$ is of the form $(x - y)^l$, e.g. the $l_1$ or $l_2$-loss we get

$$
\begin{aligned}
\mathcal{L}(f_{\theta_1} - \mathbb{E}_m[f_{\theta_m}(x)]) &= \left(f_{\theta_1} - \frac{f_{\theta_1} + f_{\theta_2}}{2}\right)^l \\
&= \frac{1}{2^l}(f_{\theta_1} - f_{\theta_2})^l.
\end{aligned}
$$

We see that the two losses are equal up to a scaling factor that disappears if we tune the coupling weight.

### E.3. Robustness of Coupled Weighting

To investigate the robustness of the coupled weighting $\gamma$, we followed the same experimental setup on CIFAR-10 with a Resnet18 model. The results can be seen in Figure 9. From the figure, we see that the validation accuracy is somewhat flat as soon as $\gamma > 1$, but there is a small optimum around $\gamma = 6$. This illustrates that at least for CIFAR-10, the choice of $\gamma$ is robust.

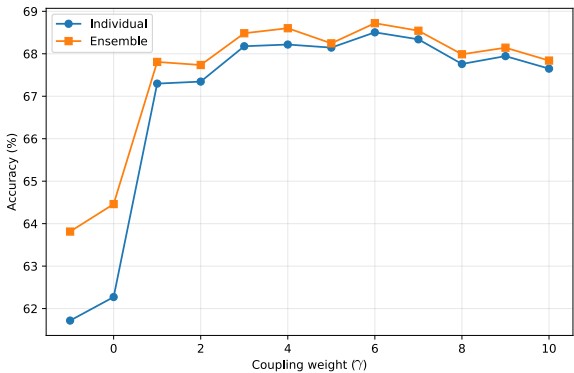

*Figure 9.* Validation accuracy as a function of the weighting of the ensemble consistency loss.

### E.4. How to Schedule the coupled loss

Initially, during training, the members of the ensemble models only have weak prediction strength. This results in the ensemble prediction serving only as a weak signal guiding the models. Intuitively, this suggests that the weighting of the coupled loss should be added or increased as training progresses. We investigate if this is the case in the same CIFAR-10 setting. We let the ensemble coupling weighting be a linear function of the number of epochs, and vary the starting value and slope of the ensemble coupling weighting. The results can be seen in Figure 10, where negative coupling weights are clipped to 0, while Figure 11 shows the un-clipped results (in the relevant area). From Figure 10, we see that for CIFAR-10, there is no large benefit to begin coupling later compared to selecting a good constant coupling value. Note that a delayed start corresponds to a negative start value and a positive increase pr. epoch, as an initial coupling of -1 and a pr. epoch increase of 0.1 means it starts at epoch 10, due to clipping.

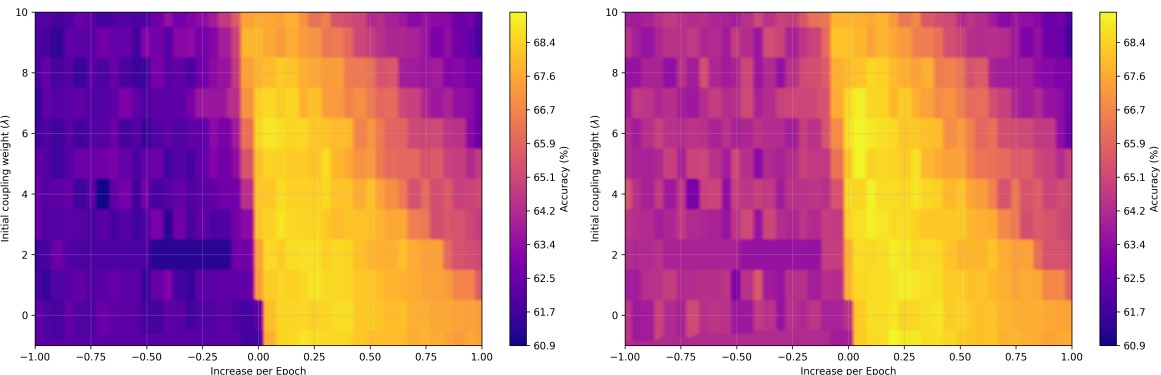

*Figure 10.* Validation accuracy as a function of the initial coupling weight and the increase in coupling weights per epoch for an individual model (left) and a coupled ensemble with two members (right). The results are averaged over 3 seeds.

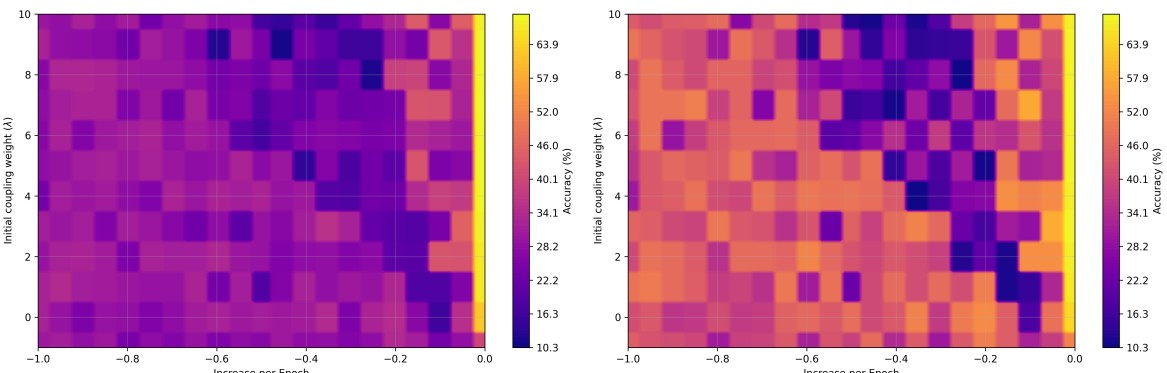

*Figure 11.* Validation accuracy as a function of the weighting of the ensemble consistency loss.

## E.5. Different Losses

We also investigated the sensitivity to different formulations of the ensemble consistency loss. The results are shown in Table 21. We ran with the same setup for the computer vision CIFAR-10 and two ensemble members. While the best performing loss function was KL-divergence (the same form as the supervised loss), the "regression" functions $(L_1, L_2, L_\infty)$ performed about the same. Only the reversed KL-divergence, $D_{KL}(E||I)$, resulted in lower accuracy, at around the same level as a decoupled model (see Table 5).

*Table 21.* Validation accuracy with different ensemble consistency loss functions. Results averaged over 10 seeds. Here, $I$ is the individual prediction and $E$ is the ensemble consensus.

| Ensemble Loss | Individual Accuracy |
|---|---|
| $L_\infty$ | $66.23_{\pm 0.29}$ |
| $D_{KL}(I||E)$ | $66.62_{\pm 0.51}$ |
| $D_{KL}(E||I)$ | $59.37_{\pm 0.78}$ |
| $L_1$ | $66.01_{\pm 0.51}$ |
| $L_2$ | $66.12_{\pm 0.45}$ |

## E.6. Different coupling strategies

We investigated different strategies for coupling the unsupervised loss on QM9. This includes various combinations of three parameters: the *coupling weight*, the *coupling start* and the *coupling schedule*.

**Coupling weight** The coupling weight parameter defines how much the unsupervised loss should contribute to the total loss. When set to 0, only the supervised loss will be taken into account.

**Coupling start** The coupling start refers to when the unsupervised loss in included during training, i.e. for the first $x\%$ of epochs, the model is only trained on the labeled data and only afterwards, the unsupervised loss with be included via coupling. Depending on the dataset and task, it intuitively can make sense to first let the model learn a little bit before evaluating the loss on unlabeled data. Specifically, in regression tasks this can be the case, since the model output is not bounded, as opposed to classification tasks. When set to 0, coupling will be used through the whole training. This parameter is given in percentage, i.e. percentage of total training epochs after which the coupling should start.

**Coupling schedule** Three different coupling schedules were tested: *constant*, *increase* and *bell*. *Constant* refers to the the coupling weight being constant from onset until the end of training. *Increase* means that the there will be a smooth ramp up until the coupling weight reaches its maximum (i.e. the coupling weight parameter). *Bell* means that there is a smooth bell curve over the coupling weight, i.e. first in increases, then decreases. Here, it will start and end at 0, and peak at a maximum which is set via the coupling weight parameter.

Figure 12 and Figure 13 shows the impact of different coupling strategies on the model performance, here for target 4 and 7 of QM9 respectively. We can see that a good choice of the coupling weight is crucial for our method to result in a significant improvement in MAE compared to the fully supervised baseline. The optimal coupling weight seems to differ per task, as both targets have a different optimum (0.1 for target 4 and 0.01 for target 7). A good value for the coupling start seems to depend on the choice of coupling weight, however a trend can be observed that for the best coupling weight options for each target, the optimal coupling start is 0, i.e. using coupling from the start of training. The optimal choice of coupling schedule seems to depend on both of the other choices, but in the specific case of target 4, the *increase* schedule led to the best performance. For target 7, the *bell* schedule resulted in the best ensemble performance, while the *constant* schedule led to the best individual performance.

One interesting finding here is that if we couple too strongly, meaning we are weighing the unsupervised loss to high, the ensemble performance gets worse than the baseline, while at the same time the individual members from the ensemble are outperforming the baseline. This is due to the models collapsing, so while each individual model is better than an individual model that was not coupled, ensembling has no significant benefit anymore.

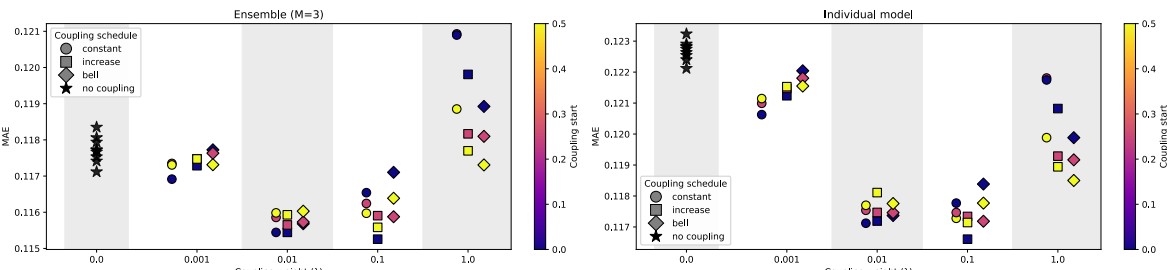

*Figure 12.* Performance (MAE) of coupled ensembles (left) and individual models from coupled ensembles (right) for different coupling strategies, for QM9 target 4.

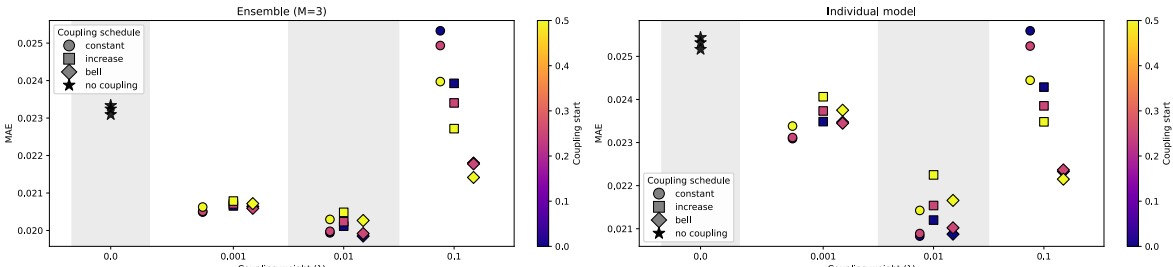

*Figure 13.* Performance (MAE) of coupled ensembles (left) and individual models from coupled ensembles (right) for different coupling strategies, for QM9 target 7.

### E.7. Evaluating Overfitting on Unlabeled Data

To evaluate potential overfitting to the unlabeled data, we compare the final model's performance on the unlabeled training set against its performance on the unseen test set. For this analysis, we leverage our access to the ground-truth labels of the unlabeled set to compute its MAE. As presented in Table 22, the performance is nearly identical across both datasets for all 12 QM9 targets. This strong correspondence indicates that our method avoids overfitting to the unlabeled data used during training. This has a significant practical benefit, as it means the model's predictions on the entire unlabeled set can be reliably used for downstream tasks.

*Table 22.* PaiNN performance (MAE) on QM9 targets, comparing the held-out test set with the unlabeled dataset used during training. Results are reported for 5 seeds.

| Target | Unit | Data | Individual Member | Ensemble (M=4) |
|---|---|---|---|---|
| $\mu$ | D | Test | $.0619_{\pm.0003}$ | $.0613_{\pm.0003}$ |
| | | Unlabeled | $.0596_{\pm.0003}$ | $.0596_{\pm.0003}$ |
| $\alpha$ | $a_0^3$ | Test | $.1322_{\pm.0011}$ | $.1303_{\pm.0011}$ |
| | | Unlabeled | $.1268_{\pm.0008}$ | $.1261_{\pm.0008}$ |
| $\epsilon_{HOMO}$ | meV | Test | $73.9789_{\pm.4368}$ | $73.0755_{\pm.4472}$ |
| | | Unlabeled | $71.7113_{\pm.4012}$ | $71.6826_{\pm.4018}$ |
| $\epsilon_{LUMO}$ | meV | Test | $57.7186_{\pm.2247}$ | $57.2369_{\pm.2159}$ |
| | | Unlabeled | $56.8810_{\pm.1844}$ | $56.8676_{\pm.1839}$ |
| $\Delta\epsilon$ | meV | Test | $117.0365_{\pm.4988}$ | $115.7195_{\pm.5100}$ |
| | | Unlabeled | $114.1592_{\pm.3078}$ | $114.1303_{\pm.3091}$ |
| $\langle R^2 \rangle$ | $a_0^2$ | Test | $.6100_{\pm.0206}$ | $.5605_{\pm.0206}$ |
| | | Unlabeled | $.5918_{\pm.0205}$ | $.5552_{\pm.0202}$ |
| ZPVE | meV | Test | $2.0138_{\pm.0054}$ | $1.9907_{\pm.0055}$ |
| | | Unlabeled | $1.9925_{\pm.0066}$ | $1.9883_{\pm.0066}$ |
| $U_0$ | meV | Test | $19.9642_{\pm.1291}$ | $19.3816_{\pm.1278}$ |
| | | Unlabeled | $19.3096_{\pm.1434}$ | $18.9715_{\pm.1416}$ |
| $U$ | meV | Test | $20.1731_{\pm.1577}$ | $19.5886_{\pm.1574}$ |
| | | Unlabeled | $19.5288_{\pm.1248}$ | $19.1908_{\pm.1234}$ |
| $H$ | meV | Test | $20.1407_{\pm.1268}$ | $19.5509_{\pm.1328}$ |
| | | Unlabeled | $19.5028_{\pm.1370}$ | $19.1620_{\pm.1355}$ |
| $G$ | meV | Test | $20.3142_{\pm.1571}$ | $19.7479_{\pm.1634}$ |
| | | Unlabeled | $19.7490_{\pm.1384}$ | $19.4296_{\pm.1400}$ |
| $C_v$ | $\frac{cal}{mol\,K}$ | Test | $.0449_{\pm.0002}$ | $.0439_{\pm.0002}$ |
| | | Unlabeled | $.0443_{\pm.0001}$ | $.0439_{\pm.0001}$ |

### E.8. Decoupling of Ensemble Gradient

Passing the gradient through the ensemble prediction could potentially lead to failure cases such as *learner collusion* (Jeffares et al., 2023). We investigate this by comparing the error under detaching and not detaching the gradient. The results are shown in Table 23 for the equivariant GNN on QM9 target $U_0$ and for the GNN architectures on peptides-struct in Table 24. We see there is no statistical significant difference between detaching and not detaching, so no major failure case is observed.

*Table 23.* The validation error on QM9 target $U_0$ with the ensemble prediction detached and attached. The error is reported as 95% standard error of the mean using 3 different seeds for each run.

|  | Individual | Ensemble |
|---|---|---|
| Detached | $0.0197_{\pm 0.0007}$ | $0.0191_{\pm 0.0006}$ |
| Not Detached | $0.0196_{\pm 0.0005}$ | $0.0190_{\pm 0.0004}$ |

*Table 24.* The test error (MAE) on peptides-struct. The error is reported as 95% standard error of the mean using 5 different seeds for each run.

| Architecture | Individual | | Ensemble | |
|---|---|---|---|---|
|  | Not Detached | Detached | Not Detached | Detached |
| GCN | $0.2870_{\pm 0.0037}$ | $0.2868_{\pm 0.0062}$ | $0.2867_{\pm 0.0038}$ | $0.2866_{\pm 0.0061}$ |
| GIN | $0.2949_{\pm 0.0045}$ | $0.2944_{\pm 0.0072}$ | $0.2943_{\pm 0.0044}$ | $0.2938_{\pm 0.0068}$ |
| GatedGCN | $0.2853_{\pm 0.0050}$ | $0.2854_{\pm 0.0061}$ | $0.2846_{\pm 0.0050}$ | $0.2848_{\pm 0.0068}$ |

## F. Computational Scaling

Training larger ensembles and coupling the ensemble increases the computational cost. We document the computational scaling for PaiNN in Table 25. The time pr. epoch was averaged over 10 epochs, and tested with a batch size of 32, on a system with 64GB memory, RTX A5000, and 4 threads of EPYC 9124.

*Table 25.* PaiNN training performance (Seconds / Epoch) for different model sizes across supervised and SSL variants.

| Ensemble Size (M) | Training (Seconds / Epoch) | | Inference (ms / Batch) |
|---|---|---|---|
|  | Decoupled | Coupled |  |
| 1 | 4.69 | - | 15.60 |
| 2 | 6.00 | 11.5 | 28.21 |
| 3 | 7.82 | 15.42 | 41.03 |
| 4 | 10.14 | 19.63 | 54.29 |
| 6 | 13.95 | - | 80.83 |
| 8 | 17.84 | - | 107.13 |

## G. Adaptive Ensemble Coupling

Early during the experimentation, we also investigated methods for adapting the coupling strength automatically. Specifically, we tried using the SoftAdapt (Heydari et al., 2019) on QM9 target ZPVE. The results can be seen in Figure 14. We observed SoftAdapt consistently found a too high coupling weight (around 0.5-2, compared to the optimal 0.01). Another potential direction could have been to use the variance within the ensemble as signal for adapting the coupling weight. The idea being if the coupling weight was too high there would be a visible phase transition in the intra-ensemble variance that could be controlled. However, we instead observed the variance decreasing smoothly (see Figure 8). This led us not to pursue this further.

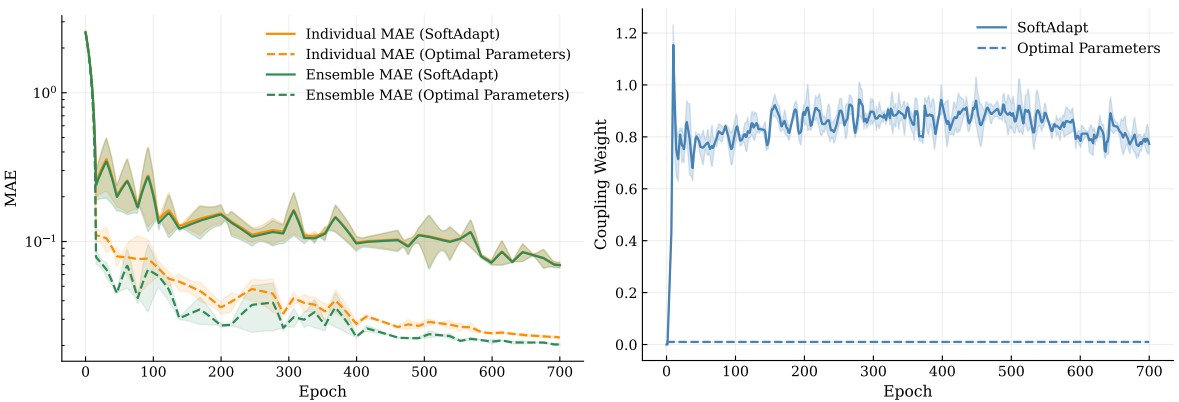

*Figure 14.* The validation error (left) and coupling weights (right) for experiments with running SoftAdapt to control the coupling strength compared against the optimal coupling strength of 0.01. Note that we only train for 700 epochs, as this parameter was not tuned at the time of these runs. The curves are averaged over 3 seeds with $95\%$ of the standard error of the mean reported

