# OpenReview forum: "Semi-Supervised Learning for Molecular Graphs via Ensemble Consensus"
_ICML.cc/2026/Conference — ICML 2026 regular_

### Official Review · Reviewer_LWd9 · 2026-03-10

**Soundness:** 3
**Presentation:** 3
**Significance:** 3
**Originality:** 3
**Overall Recommendation:** 3
**Confidence:** 3

**Summary:**

To addresses the issue of the scarcity of labeled data in molecular prediction, this paper proposes an improved semi-supervised learning method, which incorporates ensemble consensus as part of the training objective. Specifically, the paper does not perform data augmentation but, during the ensemble process, it combines the predictions of all base models for the unlabeled data as ensemble consensus and adds it to the loss function. The paper explains theoretically the rationality that ensemble methods have lower loss than independent model losses, and verifies through experiments that the semi-supervised learning (SSL) ensemble prediction method using ensemble consensus outperforms a single base model under the same training method, and is significantly superior to the ensemble method without semi-supervised learning.

**Compliance With Llm Reviewing Policy:**

Affirmed.

**Key Questions For Authors:**

1. What's the meaning of **coupling** and **decoupling** (e.g. mentioned in Table 4 in the appendix). Dose **decoupling** equal to a purely supervised model, that is, with coupling weight set to 0, or does it involve the use of other semi-supervised learning methods?
2. Supported in Appendix E.4 and E.6, authers find that there is no large benefit to begin coupling later compared to selecting a good constant coupling value. If possible, please offer some speculation about the underlying reasons.
3. How is the training data distributed? In line 388-390, it is mentioned that "the size of the training batches is doubled as it now includes both the unlabeled data as well as the labeled data." This makes me believe that B~L~ and B~U~ in Table 4 are of the same size. However, in the experimental design, it is mentioned that labeled data and unlabeled data are in a 1:9 ratio (simulating the real-world proportion), so were all the unlabeled data used in the training process?
4. How were the training, validation, and testing data for all the control experiments divided? Was only 10% of the labeled data from the training set used, or all of the training data?

**Limitations:**

yes

**Strengths And Weaknesses:**

Strengths:
1. Concise and effective method design
2. The fact that using a single model trained with the proposed SSL method outperforms a complete ensemble trained only with labeled data breaks the conventional thinking that ensemble models are superior to member models, and significantly reduces inference costs.
3. The discussion on the coupling method of supervised and SSL is very detailed, and unexpected results are obtained.
4. The experimental results have broad application significance.

Weaknesses:

1. The training volume is too large and cannot be applied to large models, which results in the unknown performance of this method on large models and it is highly likely to be inferior to that on small models.
2. There is no comparison with the state-of-the-art molecular prediction methods. Although this work focuses on improving the SSL method, a comparison with the SOTA is also necessary.

---

> ### Author Rebuttal · Authors · 2026-03-31
>
> Thank you for your time and feedback. To address your questions:
> 1. We appreciate you catched the inconsistent usage. Yes, decoupling refers to a purely supervised model (identical to setting the coupling weight to 0). We have clarified this in the submission to make it clearer.
> 2. That’s a good question. We hypothesise that early coupling regularise the models by keeping each ensemble member in more robust areas in parameter space, which helps the model learn. An alternative explanation for classification was also given by [5]. They hypothesis early coupling stabilizes learning through keeping probability mass on non-correct labels to make the models more stable. This explanation does not extend to regression settings, as it has a single head, but we see the same behavior in this setting on QM9.
> 3. In line 388-390 we are referring to the mini-batch size, as in each iteration a forward pass of the models are performed on the labeled data as well as the unlabelled data. We are indeed using all the unlabelled data. We have updated the submission to be clearer on this.
>
> We used only 10% of the labels of the training data, for the remaining 90% of training data, we did not use the label.
> To address the first weakness raised; is it possible you can elaborate on what you mean by training volume? If we understand correctly, you ask if our method extends to smaller models? Our experiments span from single models with ~13 million parameters (mol-pcba, GCN architecture) to ~112 thousand parameters (MNIST GCN architecture), so we have observed a benefit for models in these size intervals. In regards to comparison to state-of-the-art methods are you suggesting including comparison to other semi-supervised methods? To our understanding we compare against Pseudo$\sigma$ which is currently state-of-the-art for QM9 [6]. If you ask for network architectures, we can include comparisons to a fully supervised GottenNet model trained on 10% of the data. To elaborate, we have run GotenNet with their “B” hyperparameters in this setup that obtains a MAE of 0.1528 for Target $\alpha$, compared to 0.1622 MAE for fully supervised PaiNN, and still considerably higher than the 0.1322 MAE we obtain for a single model from the consensus ensemble.
>
> [5]: Zhang, Ying, et al. "Deep mutual learning." Proceedings of the IEEE conference on computer vision and pattern recognition. 2018
> [6]: Huang, Kexin, et al. "Uncertainty-aware pseudo-labeling for quantum calculations." Uncertainty in Artificial Intelligence. PMLR, 2022

---

> > ### Author Rebuttal · Reviewer_LWd9 · 2026-04-02
> >
> > I have seen additional results and clearifications.

---

> > > ### Author Response · Authors · 2026-04-08
> > >
> > > We are glad all questions and concerns have been addressed. We hope the reviewer will update their scores accordingly.

---

### Official Review · Reviewer_3HqW · 2026-03-12

**Soundness:** 2
**Presentation:** 2
**Significance:** 3
**Originality:** 2
**Overall Recommendation:** 3
**Confidence:** 5

**Summary:**

This paper proposes a semi-supervised learning approach for molecular graphs based on ensemble consensus training, where multiple models are trained jointly, and each model is encouraged to match the ensemble’s mean prediction on unlabeled data. Motivated by the classical ensemble ambiguity decomposition, the authors argue that the ensemble prediction is guaranteed to perform at least as well as the average individual model under convex losses. The method is implemented through a consistency regularization objective, that combines supervised loss on labeled data, with a consensus loss penalizing deviation from the ensemble mean on unlabeled data. The paper further shows that this consensus objective can be interpreted as a Jensen relaxation of pairwise cross-supervision methods (similar to n-CPS). Experiments on several molecular graph benchmarks and architectures demonstrate improvements over supervised baselines and selected SSL methods, and notably show that a single model trained within the consensus ensemble can outperform a standard supervised deep ensemble, although the gains are generally marginal over similar methods. Overall, the work presents a simple, theoretically motivated method with empirical validation for molecular graph learning.

**Compliance With Llm Reviewing Policy:**

Affirmed.

**Final Justification:**

My core concern about incremental novelty is only partially resolved. The domain shift from images to molecular graphs is a reasonable argument, but the method still combines existing ideas.

**Key Questions For Authors:**

Along with the weakness:
Q1 - How does the proposed method compare conceptually with approaches such as Deep Mutual Learning?
Q2 - Given that identical mini-batches are used for all models, what mechanisms maintain diversity among ensemble members? Have the authors measured prediction diversity during training?
Q3 - What practical insight should readers take from the linear vs nonlinear consensus analysis in the supplementary material? How does this analysis influence the design or behavior of the proposed method?

Answers to these questions would help clarify the novelty and strengthen the paper’s theoretical positioning.

**Limitations:**

yes

**Strengths And Weaknesses:**

Strengths:
- S1 Simple and intuitive method that seemingly avoids data augmentation, making it useful in domains such as biochem, where augmentations can drastically alter chemical properties.
- S2 The empirical evaluation appears thorough, including multiple domains, architectures, ablations, and hyperparameter tuning details.
- S3 The paper is mostly easy to follow, although several theoretical statements could be tightened to avoid overinterpretation.
- S4 Showing that a single model trained with consensus supervision can outperforms a standard ensemble on molecular GNN tasks is empirically interesting and relevant. While similar effects have been observed in distillation and Deep Mutual Learning (DML), demonstrating it convincingly on molecular benchmarks is valuable.

Weakness:
- W1. The proposed method appears only incrementally novel. Similar ensemble and cross-supervision ideas already exist, for example, n-CPS, which the authors cite but do not clearly distinguish against. Furthermore, conceptually related approaches have appeared in the computer vision literature, such as 'Deep Mutual Learning' (CVPR 2018) and collaborative knowledge distillation methods. The paper would benefit from clearer positioning and discussion of differences.
- W2 The theoretical motivation for the approach largely relies on classical ensemble ambiguity decomposition and a Taylor approximation analysis. While these arguments provide intuition, they do not appear to yield a fundamentally new theoretical insight specific to the proposed method.
- W3 The supplementary material includes two theorems analyzing consensus dynamics for linear and nonlinear models. However, their novelty and direct relevance to the proposed method are unclear. In particular, the nonlinear case appears largely heuristic and does not provide a strong theoretical guarantee.
- W4 Several theoretical claims are overstated (many of which can be fixed with better wording). For example:

> "for any convex loss function" the ensemble loss can be written as "Ensemble Loss = Average Individual Loss − Ambiguity"

This exact decomposition only holds for specific losses such as MSE (which the authors correctly show later). For general convex losses, Jensen's inequality provides an inequality rather than an exact decomposition.

> "the ensemble consensus is provably superior to the average individual model"

The theory only guarantees that the ensemble loss is not worse than the average individual loss, not that it is superior to individual models in general; the ensemble may still perform worse than the best member.

>  ensemble prediction will be a useful signal as long as the models are better than random"

This does not follow directly from the decomposition. Jensen's inequality only ensures improvement over the average loss, not that the resulting predictions are sufficiently accurate to serve as reliable training targets early in training. While experiments may support this behavior empirically, the theoretical claim should be softened in my opinion.

Moreover, the manuscript states that ensemble diversity arises from stochastic mini-batch SGD:

> "follows a distinct optimization path due to the stochastic nature of mini-batch SGD."

However, the experimental setup later states:

> "All ensemble members were trained on identical mini-batches of supervised data to simplify implementation."

Using identical mini-batches removes SGD stochasticity as a source of diversity. While different random initializations can still lead to distinct optimization trajectories, this still weakens the argument and should be clarified.

The notation is sloppy in supplementary proofs. For example:

 $O(\nabla^2_\theta)=0$

for linear models should simply be written as

 $$ \nabla^2_\theta f_\theta(x) = 0 $$

since the Hessian of a linear function is zero. The current notation is imprecise and confusing.

[Nitpick] Some statements blur hypotheses and conclusions, and should be clarified to indicate whether they represent empirical observations or theoretical conclusions, e.g.:

> "This can explain why we observe no benefit to warmup of the coupling loss."

or

> “One explaination for the small increase in accuracy is the label-smoothens assumption can be violated for graphs and especially molecules.”

These appear to be hypotheses rather than derived results.

The paper claims decreases in gradient norm and Hessian trace are "predicted by the theory" and imply robustness, but the theoretical analysis does not formally establish this relationship; Figure 3 mainly shows a regularization effect rather than confirming the stated theory.
Smaller coupling seem to better.

Overall, the argument that the ensemble's theoretical properties justify its use as a consistency target is reasonable. However, the theoretical result only establishes that the ensemble improves upon the average model, not that it necessarily provides high-quality pseudo-labels during early training. The manuscript also tends to overstate the claim that ensemble predictions are always more accurate. In general, there is no guarantee that unlabeled targets produced by the ensemble constitute high-quality supervisory signals, particularly early in training.

---

> ### Author Rebuttal · Authors · 2026-03-31
>
> We greatly appreciate the detailed feedback, especially on the specific formulation and notational issues. We agree some of the formulations should be restricted and have changed the work. The submission is more clear now.
>
> 1. The conceptual idea of the two methods are similar. Booth methods use the intuition that multiple models learn from each other to further bootstrap learning. Where the methods conceptually differ is how they interact and the domain used. Deep Mutual Learning (DML) is designed for fully-supervised tasks where the intuition is that mutual learning smooths the learning process by retaining probability mass on non-true labels. Conceptually, this idea does not extend to regression tasks, but it is mathematically close. Mathematically, the change is average over scalar values instead of probabilities, change the loss function from KL-divergence and add a coupling weight. However, from their framework it is not clear why that would benefit regression. Our work employs the perspective from ensemble literature that naturally works for booth regression and classification. DML is certainly related to our work that has been added to the work; we thank the reviewer for the suggestion. We argue the main contribution is the novel and extensive testing on graph based data with a focus on molecular data. This difference also applies to N-CPS, which like DML also only investigates image based data such as CIFAR10. Image-based and graph-based modalities are very different in semi-supervised settings due to violations of smoothness and clustering assumptions. That is one of the reasons why more recent image-based semi-supervised methods can achieve comparable performance to even fully supervised methods with very limited data (100 samples from CIFAR10). Crucially, these methods do not extend to graph based data. Finally, while not the main focus of our work, we also include investigations into how ensemble consus impacts the calibration of the models, as this is often one of the main concerns in settings where ensembles traditionally are used. This is not investigated by DML or N-CPS.
>
>
> 2. Thank you for pointing out the unclear argument. What we meant to say is that the intra-ensemble variance comes from the “chaotic” nature of the learning trajectory during training Specifically, different parameter initialisations, the non-infinitesimal steps during gradient updates, and not full batch gradient descent [4].  Based on the findings of Figure S4 in Appendix B of [4] different initialisations and learning rates were the main contributions to intra-ensemble variance, so we chose to reduce computational overhead and simplicity by removing the different mini-batches for the members. We have also added a plot of the intra-ensemble variance during training (see comment for reviewer 61mR).
>
> 3. A hypothesis for the observed error reduction in the coupling of ensembles simply increases the effective size of a decoupled ensemble. This implies in the limit $M\rightarrow \infty$, that the coupled ensemble prediction converges to the mean of the decoupled ensemble. The analysis indeed shows this to be the case for linear models as the ensemble prediction does not change after a coupling gradient update. The limit of the analysis is it does not guarantee linear models cannot converge to the same solution during full training, only for a single update. We have extended the analysis to include this.
>
> This hypothesis is no longer true for non-linear models, as the coupled update can change the prediction of the full ensemble. While this does not guarantee that the coupled ensemble cannot converge to the same solution as that of the decoupled ensembles, just that there is a mechanism that enables different convergence of the coupled ensemble. The analysis does show that the coupled ensemble will have a preference to move towards flatter minima, suggesting a different solution than the decoupled ensemble. We argue this contribution is notable, as it explains why we see a benefit for coupled ensembles in both classification and regression settings. We also support this predicted behavior of flatter minima with experimental observations in Figure 1 (right).
>
> This provides the practical insight that the preference for flatter loss landscapes imply that coupling the ensemble has a regularising effect on the individual model. We used this intuition when designing the experiments to rule out that the error reduction observed was simply due to the individual models being under-reguralised through too low weight decay or too high learning rates. This was one of the reasons we re-tuned specifically the learning rate and weight decay values to rule this out. Conversely, this also suggests better performance of the coupled ensemble can be achieved by changing specifically these parameters.
>
> [4]: Stanislav, Huiyi Hu, and Balaji Lakshminarayanan. "Deep ensembles: A loss landscape perspective." arXiv preprint arXiv:1912.02757 (2019)

---

> > ### Author Rebuttal · Reviewer_3HqW · 2026-04-03
> >
> > The rebuttal addresses the main issues I raised about theoretical positioning and the relation to prior work, and I appreciate the clarifications and planned changes.
> >
> > On novelty, the authors are upfront that the conceptual idea is close to DML and n-CPS, but make a reasonable case for what is different: the focus on molecular graphs, the grounding in ambiguity decomposition, and the calibration analysis which neither DML nor n-CPS cover.
> >
> > On the theory, the nonlinear analysis is now framed more honestly as showing a preference for flatter minima rather than guaranteeing convergence, and connecting it to the deep ensembles literature helps. I no longer think it is being over-interpreted.
> >
> > On diversity, the clarification about what "stochastic SGD" meant here is fair, and assuming the authors will add  the new diversity plot directly answers what I asked.
> >
> > That said, my core concern about incremental novelty is only partially resolved. The domain shift from images to molecular graphs is a reasonable argument, but the method still combines existing ideas. I am moving from reject to weak reject.

---

> > > ### Author Response · Authors · 2026-04-08
> > >
> > > We are glad we are in agreement about the theoretical conclusions now. In regards to the plot about intra-ensemble variance, we have uploaded experiments here https://osf.io/9m46f/overview?view_only=e38c093058134bc182113a5147bc7e02, The figures for the intra-ensemble variance can be found in the files ensemble_variance_epochs_QM9_unsupervised.svg and ensemble_variance_epochs_QM9_validation.svg, for unsupervised and validation data respectively. Note the figures are very similar. In regards to novelty of the work, we argue that, while the conceptual idea of ensemble consensus is adjacent to existing work, the comprehensive evaluation demonstrates the concept’s applicability in an underexplored modality (molecules), to show it outperforms existing work in this area while also providing new theoretical insights. We argue this aligns well with ICML’s novelty criteria  of  “... rather, a work that provides novel insights by evaluating existing methods, or demonstrates improved understanding is also equally valuable” as expressed here “https://icml.cc/Conferences/2026/PeerReviewFAQ”. Once again we appreciate the reviewers thorough suggestions and questions.

---

### Official Review · Reviewer_61mR · 2026-03-12

**Soundness:** 2
**Presentation:** 2
**Significance:** 2
**Originality:** 3
**Overall Recommendation:** 4
**Confidence:** 3

**Summary:**

This paper proposes a way to improve molecular property prediction models when you have very few labeled molecules but many unlabeled ones. The idea: train multiple copies of the same model with different random starting points. On unlabeled molecules, average all models' predictions and use that average as a teaching signal to push each individual model to agree with the group. The math guarantees that the group average is at least as accurate as any individual model on average, so this creates a "better-than-self" learning signal. The paper shows this works across many molecular datasets, multiple GNN architectures, and even a realistic drug discovery task, consistently beating models trained on labeled data alone.

**Compliance With Llm Reviewing Policy:**

Affirmed.

**Final Justification:**

The authors have sufficiently provided responses to my queries around experimentation and hence have updated the scores.

**Key Questions For Authors:**

* Can you provide results across a range of label fractions (1%, 5%, 10%, 25%, 50%, 100%)? At what point does the benefit of unlabeled data become negligible? Is there a regime where the method actually hurts performance?
* What happens when the unlabeled data comes from a different distribution than the labeled data? Pick an affinity dataset like Polaris benchmark with scaffold split and buitna-clustering based splitting to evaluate the OOD performance.
*  Can you report wall-clock training times for the ensemble consensus method versus the supervised ensemble baseline and a single model? What is the actual compute overhead, accounting for the forward passes needed on unlabeled data?
* How does the method compare to recentself-supervised pre-training methods (e.g., pre-train on unlabeled molecules with a self-supervised objective, then fine-tune on labeled data)? These methods also leverage unlabeled data but through a different mechanism.
* Can you provide quantitative measurements of ensemble diversity (e.g., average pairwise prediction disagreement or the "ambiguity" term from your decomposition) throughout training? Does diversity decrease over training, and if so, does this correlate with diminishing returns from the consistency loss?
* Does the method's benefit continue to increase with larger ensembles (M=8, 16) on the GNN+ benchmarks? Is there a point of diminishing returns, and does the optimal M depend on the task or dataset size?

**Limitations:**

yes

**Strengths And Weaknesses:**

Strengths

- Overall, the paper is well written and motivation is clearly explained with sufficient background.
- The use of the ensemble ambiguity decomposition to justify the consensus target is novel. It provides a clear reason why the ensemble mean is a superior pseudo-label compared to self-training or Mean Teacher approaches.
- The formulation explicitly detaches gradients through the consensus, mitigating issues like learner collusion and aligning the optimization with an intuitive distillation view.
- The method requires no molecular-specific augmentations, no complex training schedules, and minimal hyperparameter tuning (primarily λ). It can be applied as a drop-in addition to any existing ensemble training pipeline.


Weakness

* The paper fixes the labeled fraction at 10% throughout. It is unclear how the method behaves at different label ratios ( 1%, 5%, 15%,20%, 50%). The benefit likely diminishes as more labeled data is available, but the crossover point is unknown. Similarly, the method's behavior in extremely low-label regimes (1-2%)
* The unlabeled 90% is drawn from the same dataset distribution as the labeled 10%. In real drug discovery, unlabeled molecules often come from different chemical scaffolds with distribution shift. The paper does not evaluate robustness to distribution mismatch between labeled and unlabeled pools.
* Training M=4 models costs roughly 4× the compute and memory of a single model. While the paper argues that individual models reach ensemble-level performance (enabling single-model inference), the training cost is still substantial. A detailed wall-clock time comparison and discussion of compute-performance tradeoffs is missing. Also, the paper studies M with {1,2,3,4} on a single QM9 target, this analysis is not extended to other datasets. The optimal M may vary across tasks and architectures, and larger ensembles (M=8, 16) are not explored.

---

> ### Author Rebuttal · Authors · 2026-03-31
>
> Thank you for your time and helpful feedback.
> To answer your questions in order:
> 1. Thank you for the suggestion, we have added an experiment with data fractions of $1\%, 5\% 10\%, 25\%, 50\%, 75\%$, and $100\%$ of labelled to unlabelled data for QM9 ZPVE (target 7). For these data fractions we do not find a collapse in the SSL setting. There is a consistent benefit across all splits, with the benefit narrowing for the more label abundant fractions as expected.
> Reviewer fZAp also asked for experiments to investigate out-of-distribution data. To not further complicate the setup with another dataset suite, we have implemented scaffold splitting with Butina-clustering for QM9. Specifically, we split QM9 into two distributions for the unsupervised data and one for the rest. We have run this setup for 5 seeds for ZPVE, and find that individual MAE for the supervised-only setup goes from $23.27 \pm 0.023$ to $19.60 \pm 0.026$ for the consensus trained models, and $20.18 \pm 0.0022$ to $19.02 \pm 0.027$ for the ensemble. We also find this to be an important addition, and are continuing this direction for more targets as well. While scaffolding does not yield data from highly different distributions, we think this experiment together with the experiments using the dataset PCQM4Mv2 should illuminate the SSL setup in regards to robustness in when unlabelled data is out of distribution from the training data.
> 2. Thanks for the suggestion of adding measured wall time. We have added a table of measured wall time together with a plot that shows error versus wall time. In addition, we are rewriting that part of the discussion to make it clearer. The takeaway is that the wall time closely follows the O(2M) scaling including forward passes on the unlabelled data.
> 3. As mentioned in the comment for reviewer fZAp, we are investigating this setup, but the experiments are still running.
> 4. We have added a plot to the submission with the ensemble diversity (specifically variance within the ensemble) over training runs for different coupling strengths (1, 0.1, 0.01, 0.001, 0.0001, 0). From this new plot and the old plot of Figure 10, we do observe that the intra-ensemble variance is reduced throughout training for all coupling strengths (including no coupling) as you are suggesting. Higher coupling strengths result in lower intra-ensemble variances.
> More specifically, this reduction in ensemble ambiguity does correlate with a reduction in validation error, although this conclusion is slightly muddied by the models also still learning from the supervised signal. Focusing on the individual error of the runs with a coupling strength of $(10^{-3}, 10^{-4}, 0 )$, we see their error vs intra-ensemble variance data points roughly trace out a trend. Here, the runs with larger coupling strength reduce intra-ensemble variance while reducing the individual error. Notably, the ensemble setup observed to have the lowest intra-ensemble variance (last epoch with coupling strength 1) does not have the lowest error, highlighting the trade-off between balancing ambiguity and bias within the ensemble.  What is not explained by the bias versus ambiguity is that the error of the ensemble also can be reduced with higher coupling strength for lower intra-ensemble variance
> 5. We strongly expect the improvement of going from M to M+1 ensemble members will be smaller i.e. diminishing returns. This is based on the results of standard (decoupled) ensemble literature and observed across our experiments of scaling to 32 members (although with an CNN based architecture). What is the optimal ensemble size depends on how much additional compute you would use for this. Based on our findings, a notable ensemble size is for $M=8$ and upwards. From Table 4 (beginning at line 810), we see that the increase in accuracy of the both coupled and decoupled ensembles are marginal for ensemble sizes 8 and up. In addition, this is also when the calibration of the decoupled ensemble reaches that of the coupled ensemble based on NLL (Table 5, line 834) and AUC-ROC (Table 6, line 851). We do not have the resources to investigate large ensemble sizes of 8 or 16 for all GNN+ benchmarks as the GNN models used are high in parameter count. We have instead added an experiment on the dataset Peptides-struct for the GCN architecture for ensemble sizes of 1,2, 3, 4, 6, 8, and 16 (last not finished). We generally observe the decoupled ensemble not scaling as well here as QM9 and especially CIFAR10 (image). As a result we do not see any improvement for the coupled ensemble beyond $M=3$.

---

> > ### Author Rebuttal · Reviewer_61mR · 2026-04-03
> >
> > Thank you for the responses. Some of my queries are resolved. However, as some of my questions 2 and 4 are under still experimentation, I will wait to hear back.

---

> > > ### Author Response · Authors · 2026-04-08
> > >
> > > To save space and show plots we have uploaded (anonymously) the results of the experiments on OSF https://osf.io/9m46f/overview?view_only=e38c093058134bc182113a5147bc7e02.
> > >  The results for each of the experiments:
> > > 1. We visualise the compute scaling two ways. We have included the error plotted against inference time (error_vs_inference_time.svg) and training time (error_vs_training_time.svg) for the different configurations. The error is MAE in meV with seconds pr. Epoch on QM9 for target ZPVE. Additional details are: one epoch here is defined to mirror our QM9 Experiments, i.e. $10\%$ of the full QM9 supervised data, while the SSL models also iterates over an equal amount of unlabelled batches. From the error vs training compute graph, that training time is both impacted by the number of ensemble members. Due to the ensemble members only communicating their predictions, the ensemble size can largely be mitigated by distributing the ensemble members across different devices and collecting predictions with all_gather/all_reduce. We also see that the coupled ensembles are roughly 2x slower than the decoupled training, as we simply have more unlabelled data to iterate over. From the error vs inference, we see how a single model from the SSL ensemble performs much better than a full normal ensemble at a fraction of the inference time.The details are: we used a batch size of 32 for training and 256 for inference speed with warmup, all on a single NVIDIA RTX A5000, and averaged over 10 epochs or 50 batches respectively. We used Pytorch version 2.7.1 and CUDA 11.8. We also include a table with the timings here:
> > >
> > > 2. The figures for the intra-ensemble variance can be found in the OSF files ensemble_variance_epochs_QM9_unsupervised.svg on unsupervised data and ensemble_variance_epochs_QM9_validation.svg. Note the figures are very similar.
> > > 3. For completion, we have also uploaded the plots to the error versus different labeling fractions under the name individual_MAE_vs_labelled_frac.svg.
> > > 4. In regards to the unsupervised results,  as previously mentioned we experimented against the unsupervised method Frad. We generally find that the coupling has a negative effect when used together with the unsupervised loss. For target $\epsilon_{\text{homo}}$ we see an individual MAE of 16.8985 with ensemble consensus versus 15.4895 unsupervised only. We hypothesise this could be due to two reasons. Firstly, that the dataset PCQM9 is too far away from QM9 for the ensemble consensus to be highly effective. Secondly, that the model having to agree on the ensemble consensus loss and the unsupervised target on the same data causes instabilities, as we see the models training trajectory being highly unstable.
> > > 5. The second hypothesis is supported by further experiments with Butina scaffolding on QM9. The results are in file “table_QM9_Butina.png” on OSF.  For all targets completed (target 0 to 8) we see an improvement from ensemble consensus. This suggests ensemble coupling is effective when the unlabelled distribution is not too far away from the labelled distribution, as was hypothesised by reviewer fZAp, but PCQM9 is too far out-of-distribution. We have added this finding to the Discussion section, highlighting this as a future direction as we can conclude that 3D positional unsupervised methods are preferred for data that is out-of-distribution. Note, that the non-consensus runs for target 0 perform poorly, presumably due to the distributional shift being large enough to destabilise the runs.
> > > 6. We also tried combining the ensemble with two (different seeds) pretrained Frad models. We observe a small improvement for target $\epsilon_{\text{homo}}$ with 17.5332 coupled versus 17.01439 decoupled.  This is surprising, as it suggests the coupled ensembles can still extract some information from the unlabelled data, as the models were pretrained on the same data.
> > >
> > > We thank the reviewer for the suggestions. We believe that the new experiments together with the previously requested experiments have improved the work and hope the reviewer’s questions have been addressed. If so, we hope the reviewer will update the scores accordingly.

---

### Official Review · Reviewer_fZAp · 2026-03-12

**Soundness:** 3
**Presentation:** 3
**Significance:** 4
**Originality:** 3
**Overall Recommendation:** 6
**Confidence:** 4

**Summary:**

This paper proposes a semi-supervised learning (SSL) method for molecular graph property prediction that avoids reliance on label-preserving data augmentations, a notoriously difficult requirement in the molecular domain. Instead, the method couples an ensemble of GNNs through a consensus loss: each member is trained on labeled data with a standard supervised loss and simultaneously pulled toward the ensemble's aggregate prediction on unlabeled data. The approach is grounded in the bias-variance (ambiguity) decomposition of ensemble error, which theoretically justifies the ensemble consensus as a higher-quality supervisory signal than any individual member. Empirically, the method is evaluated on QM9 (regression, PaiNN), a diverse GNN+ benchmark suite (GCN, GIN, GatedGCN), CIFAR-10, and several non-molecular graph datasets. A notable finding is that a single member from the consensus-trained ensemble consistently outperforms a full standard (decoupled) ensemble, and the method also reduces calibration error for individual models.

**Compliance With Llm Reviewing Policy:**

Affirmed.

**Final Justification:**

The authors are able to address all questions and concerns that the reviewer had. Score adjusted to 6.

**Key Questions For Authors:**

1. Can the authors try some pretrained GNN baselines (e.g., AttrMasking, ContextPred, or similar)? It would be interesting to compare the proposed framework with one of those.
2. The reviewer is interested in learning about the robustness of the framework when dealing with out-of-distribution data. The authors don't need to show good results as this is super challenging, but seeing the gap would be sufficient.
3. Can the coupling weight be adaptive instead of doing expensive sweep?
4. Appendix E.9 shows no statistical difference between detaching and not detaching the ensemble gradient. However, is this result consistent across datasets and architectures, or is it specific to QM9?
5. Can the authors clarify what drives the inconsistency in full-ensemble calibration between the molecular (ogbg-molhiv) and vision (CIFAR-10) settings? Is this an artifact of the coupling weight not being tuned for calibration, or a more fundamental problem?

**Limitations:**

1. The O(2M) training cost, which makes the method less practical for large-scale foundation model settings.
2. Reliance on per-task $\gamma$
3. Out-of-distribution is challenging given that the framework is not designed to tackle this issue.

**Strengths And Weaknesses:**

Strengths:
1. The problem is well-motivated. The paper clearly identifies a genuine gap: mainstream SSL methods are poorly suited to molecular data due to the lack of valid augmentations. The ensemble consensus approach elegantly sidesteps this constraint.
2. The theoretical grounding is solid and the use of Krogh-Vedelsby ambiguity decomposition is clean. The authors also add meaningful depth by extending to cross-entropy and nonlinear model analysis in Section 4.3 and Appendix A.
3. Strong empirical results and comprehensive ablations.

Weaknesses:
1. The reviewer thinks this paper can be further corroborated by comparing with self-supervised and transfer learning baselines. The authors acknowledge this gap but do not address it. Given that pretraining on large unlabeled molecular databases is a dominant paradigm in molecular ML, bringing these baselines into the picture can significantly improve the value of the paper.
2. The use of artificial semi-supervised splits is slightly overlooked. While common in SSL benchmarks, this conflates in-distribution unlabeled data with the real-world scenario where unlabeled data may come from a different distribution than the labeled set. The reviewer understands that this problem is very challenging, so this weakness is just for discussion and does not affect the overall review.
3. The paper notes that high coupling weights can cause ensemble to collapse, eliminating ensemble benefit. However, there is no systematic approach or diagnostic offered to detect or prevent this in practice.

---

> ### Author Rebuttal · Authors · 2026-03-31
>
> We greatly appreciate the time and specific feedback received. To respond to the respective questions:
> 1. Question 1-2: Thanks for the great suggestions. Reviewer 61mR also expression interest in unsupervised pretrained models so we are adding an experiment that compares our approach against unsupervised models and together (using the unsupervised model as a pretrained model). We chose to use Frad [1] as it a is unsupervised model specifically designed for 3D molecular datasets such as QM9. The experiments are conducted using the dataset PCQM4Mv2 [3] as unlabelled data and the whole of QM9 as labeled data. These experiments are still running, so we cannot comment on the results for this at the current time.
> To further investigate coupled ensemble's robustness to shifts in ada distributions, we have also investigated the results with more difficult scaffolding as suggested by reviewer 61mR. While scaffolding does not yield data splits of highly different distributions, we still observe an clear improvement in this setting (see the comment for reviewer 61mR) . This suggests coupling ensembles have some robustness when the data distributions are not identical.
> 2. Question 3: Thank you for the suggestion. We have added a section in the appendix describing a failed attempt we conducted at automatic or adaptive selection of the coupling strength. Specifically, we tried using the SoftAdapt method [3] on QM9 target ZPVE. We observed SoftAdapt consistently found a  too high coupling weight (around 0.5-2, compared to the optimal 0.001). We had hoped we could use the variance within the ensemble as a signal for adapting the coupling weight, as we expected there would be some kind of transition in the variance as the coupling became too high. However, we instead observed the variance decreasing smoothly. This made us not chase this idea further. Any ideas for this would be welcome.
> 3. Question 4: Thank you for the suggestion, we have added this experiment for the Peptides-Struct dataset for all three architectures across 5 seeds. We find the difference in MAE to not be statistically different when detaching and not detaching for any of the three architectures. This has been added to the submission.
> 4. Question 5: We hypothesize the difference of the full ensemble between the two datasets is due to saturation of the models. For booth datasets we see a calibration benefit of the coupled ensembles for $M\leq 4$, it is only for the larger ensemble sizes that the benefit for CIFAR10 (image) is reduced. In regards to the worse ECE observed score for CIFAR10 (image), we speculate this could be due to the classes in the dataset being heavily clustered, resulting in the coupled ensembles having sharp decision boundaries between them. This could result in few samples being confidently wrong, thereby increasing the more sensitive ECE score while improving LL and ROC-AUC. We hypothesise a lower coupling weights would alleviate this problem.
>
> [1]: Feng, Shikun, et al. "Fractional denoising for 3d molecular pre-training." International Conference on Machine Learning. PMLR, 2023
>
> [2]: Nakata, Maho, and Tomomi Shimazaki. "PubChemQC project: a large-scale first-principles electronic structure database for data-driven chemistry." Journal of chemical information and modeling 57.6 (2017): 1300-1308.
>
> [3]: Heydari, A. Ali, Craig A. Thompson, and Asif Mehmood. "Softadapt: Techniques for adaptive loss weighting of neural networks with multi-part loss functions." arXiv preprint arXiv:1912.12355 (2019)

---

> > ### Author Rebuttal · Reviewer_fZAp · 2026-04-02
> >
> > Thank you the authors for providing additional results and clarification. These completely clarify the questions and concerns that the reviewer had. The reviewer decides to strongly support this paper for acceptance. Score has been changed to 6. Good luck!

---

> > > ### Author Response · Authors · 2026-04-08
> > >
> > > We are happy that the reviewer found the work novel and comprehensive. We greatly thank the reviewer for suggestions and feedback for experiments.

---

### Decision · Program_Chairs · 2026-04-30

**Decision:**

Accept (regular)

**Comment:**

This paper proposes an ensemble consensus SSL method for molecular graph property prediction, where each member is trained to match the ensemble aggregate on unlabeled data. Reviewers broadly acknowledge the clear motivation and that the method elegantly avoids molecular augmentations (fZAp, 61mR, 3HqW), and three find the single-model-beats-decoupled-ensemble result genuinely interesting (fZAp, LWd9, 3HqW). Main concerns are incremental novelty relative to n-CPS and Deep Mutual Learning (3HqW), limited label-fraction and OOD evaluation (fZAp, 61mR), and missing self-supervised pretraining baselines (fZAp, 61mR). The rebuttal adds Butina scaffold splits showing consistent gains, varied labeling fractions, wall-clock timings, intra-ensemble variance plots, and a partial Frad comparison. fZAp raised the score to 6 and LWd9 marks concerns fully resolved; 61mR and 3HqW settle on partially resolved, with 3HqW accepting the domain-shift argument but still flagging incremental novelty. The remaining issues are real but does not outweigh the breadth of evidence and the surprising single-model finding. On balance, the contributions are strong enough to justify acceptance.